# Neural Characteristic Activation Analysis and Geometric Parameterization for ReLU Networks

**Wenlin Chen**
University of Cambridge
MPI for Intelligent Systems
wc337@cam.ac.uk

**Hong Ge**
University of Cambridge
hg344@cam.ac.uk

## Abstract

We introduce a novel approach for analyzing the training dynamics of ReLU networks by examining the *characteristic activation boundaries* of individual ReLU neurons. Our proposed analysis reveals a critical instability in common neural network parameterizations and normalizations during stochastic optimization, which impedes fast convergence and hurts generalization performance. Addressing this, we propose *Geometric Parameterization (GmP)*, a novel neural network parameterization technique that effectively separates the radial and angular components of weights in the hyperspherical coordinate system. We show theoretically that GmP resolves the aforementioned instability issue. We report empirical results on various models and benchmarks to verify GmP's advantages of optimization stability, convergence speed and generalization performance.

## 1 Introduction

In standard neural networks, each neuron performs an affine transformation on its input $\mathbf{x} \in \mathbb{R}^n$ followed by an element-wise nonlinear activation function $g$:

$$z = g(\mathbf{w}^{\mathrm{T}} \mathbf{x} + b), \tag{1}$$

where the affine transformation is parameterized by a weight vector $\mathbf{w} \in \mathbb{R}^n$ and a bias scalar $b \in \mathbb{R}$. Rectified Linear Unit (ReLU) [17] is arguably the most popular activation function in modern deep learning architectures due to its simplicity and effectiveness:

$$g(s) = \mathrm{ReLU}(s) = \max(0, s). \tag{2}$$

We introduce a novel concept for ReLU networks, called *characteristic activation boundary*, which is defined as the set of input locations with zero pre-activations. By definition, such boundaries separate the active and inactive regions of ReLU neurons in the input space, which play a critical role in ReLU networks since they serve as the fundamental building blocks for the decision boundaries.

In this work, we analyze the evolution dynamics of the characteristic activation boundaries in ReLU networks. Our novel analysis identifies a critical instability in many common parameterizations and normalizations that operate in the Cartisian coordinate, including Standard Parameterization, Weight Normalization [56], Batch Normalization [24] and Layer Normalization [2]. We show theoretically that this issue destabilizes the evolution of the characteristic boundaries in the presence of stochastic gradient noise, and empirically that it impedes fast convergence and hurts generalization performance. To address this, we introduce a novel neural network parameterization, named *Geometric Parameterization (GmP)*. As opposed to traditional parameterizations and normalizations which operate in the Cartesian coordinate, GmP operates in the hyperspherical coordinate which automatically decouples the radial and angular components of the weights. Our theoretical results show that GmP stabilizes the evolution of characteristic activation boundaries during stochastic optimization. Our empirical

38th Conference on Neural Information Processing Systems (NeurIPS 2024).

studies confirm the efficacy of GmP on various models and benchmarks. We report notable empirical improvements in optimization stability, convergence speed, and generalization performance, which validate our hypotheses and theoretical results.

## 2 Neural Characteristic Activation Analysis for ReLU Networks

This section defines characteristic activation boundary and its geometric connection to ReLU features, which serve as the basics of the proposed analysis for understanding neural network training dynamics.

### 2.1 Preliminary and Terminology

**Standard Parameterization (SP)** refers to the weight-bias parameterization as defined in Eq (1).

**Weight Normalization (WN)** [56] is a reparameterization technique that decouples the length $l$ and the direction $\mathbf{v}/\|\mathbf{v}\|_2$ of $\mathbf{w}$ in a standard ReLU unit (1):

$$z = \text{ReLU}\left(l\left(\frac{\mathbf{v}}{\|\mathbf{v}\|_2}\right)^{\text{T}}\mathbf{x} + b\right).$$

(3)

WN makes the length $l$ and the direction $\mathbf{v}/\|\mathbf{v}\|_2$ of the weight vector independent of each other in the Cartesian coordinate system, which is effective in improving the conditioning of the gradients and thus speeding up optimization.

**Batch Normalization (BN)** [24] is a widely-used normalization layer in modern deep learning architectures such as ResNet [22], which is effective in accelerating and stabilizing stochastic optimization of neural networks [29]. BN standardizes the pre-activation using the empirical mean and variance estimated by mini-batch statistics:

$$\text{BN}(\mathbf{w}^{\text{T}}\mathbf{x} + b) = \gamma\frac{\mathbf{w}^{\text{T}}\mathbf{x} - \hat{\mathbb{E}}_{\mathbf{x}}[\mathbf{w}^{\text{T}}\mathbf{x}]}{\sqrt{\hat{\text{Var}}_{\mathbf{x}}[\mathbf{w}^{\text{T}}\mathbf{x} + b]}} + \beta,$$

(4)

where $\gamma, \beta \in \mathbb{R}$ are two free parameters to be learned from data, which adjusts the output of the BN layer as needed to increase its expressiveness.

### 2.2 ReLU Characteristic Activation Boundary

Noticing that the ReLU activation function (2) is active for positive arguments $s > 0$ and inactive for negative arguments $s < 0$, we introduce a novel concept called *characteristic activation boundary (CAB)* at the cut-off point $s = 0$ (i.e., the point with zero pre-activations), which will play a central role in our proposed characteristic activation analysis.

**Definition 2.1** (CAB)**.** The *characteristic activation boundary (CAB)* for a ReLU unit is defined by the set of input locations with zero pre-activations:

$$\mathcal{B}(\mathbf{w}, b) = \{\mathbf{x} \in \mathbb{R}^n : \mathbf{w}^{\text{T}}\mathbf{x} + b = 0\}.$$

(5)

A CAB is an $(n-1)$-dimensional hyperplane that separates the active and inactive regions of a ReLU unit in the input space $\mathbb{R}^n$. Fig 1(a) visualizes a CAB in $\mathbb{R}^2$.

**Definition 2.2** (Spatial location)**.** The *spatial location* of a CAB is defined as

$$\phi(\mathbf{w}, b) = -\frac{b\,\mathbf{w}}{\mathbf{w}^{\text{T}}\mathbf{w}} = -\frac{b}{\|\mathbf{w}\|_2}\frac{\mathbf{w}}{\|\mathbf{w}\|_2},$$

(6)

which is a point on the corresponding CAB since $\mathbf{w}^{\text{T}}\phi + b = 0$. Not that the vector that goes from the origin to $\phi$ specifies the shortest path between the origin and the CAB. Therefore, each spatial location uniquely determines a CAB. Fig 1(a) visualizes the spatial location of a CAB in $\mathbb{R}^2$. CABs play a critical role in ReLU networks, since they effectively specify the locations of ReLU features (i.e., non-linearities) which serve as the building blocks for the decision boundaries.

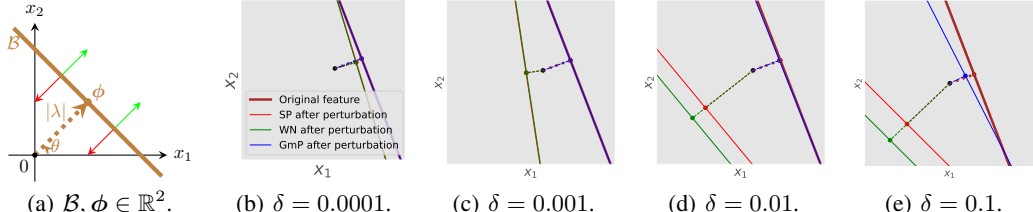

(a) $\mathcal{B}, \phi \in \mathbb{R}^2$.     (b) $\delta = 0.0001$.     (c) $\delta = 0.001$.     (d) $\delta = 0.01$.     (e) $\delta = 0.1$.

Figure 1: (a) Characteristic activation boundary (CAB) $\mathcal{B}$ (brown solid line) and spatial location $\phi = -\lambda \mathbf{u}(\theta)$ of a ReLU unit $z = \text{ReLU}(\mathbf{u}(\theta)^\text{T} \mathbf{x} + \lambda) = \text{ReLU}(\cos(\theta)x_1 + \sin(\theta)x_2 + \lambda)$ for inputs $\mathbf{x} \in \mathbb{R}^2$. The CAB forms a line in $\mathbb{R}^2$, which acts as a boundary separating inputs into two regions. Green arrows denote the active region, and red arrows denote the inactive region. (b)-(e) Stability of the CAB of a ReLU unit in $\mathbb{R}^2$ under small perturbations $\varepsilon = \delta \mathbf{1}$ to the parameters. Solid lines denote characteristic activation boundaries $\mathcal{B}$, and colored dotted lines connect the origin and spatial locations $\phi$ of $\mathcal{B}$. Smaller changes between the perturbed and original boundaries imply higher stability. GmP is most stable against perturbations.

## 2.3 Instability of Characteristic Activation Boundary During Stochastic Optimization

In the presence of stochastic gradient noise, we identify an instability issue in the evolution of CABs under common neural network parameterizations and normalizations.

**Proposition 2.3** (Instability of SP). *A perturbation $\varepsilon$ to the weight $\mathbf{w}$ under SP (1) can result in an arbitrarily large change in the angular direction of the CAB if $\mathbf{w}$ has a similar magnitude to $\varepsilon$[1].*

*Proof.* The change in the angular direction of the CAB under SP is given by

$$\langle \mathbf{w}, \mathbf{w} + \varepsilon \rangle \equiv \arccos \left( \frac{\mathbf{w}^\text{T}(\mathbf{w} + \varepsilon)}{\|\mathbf{w}\|_2 \|\mathbf{w} + \varepsilon\|_2} \right). \tag{7}$$

Since $\mathbf{w}$ has a similar magnitude to $\varepsilon$, in the extreme case one can construct a small perturbation $\varepsilon = -(1 + \delta) \mathbf{w}$ with an infinitesimal $\delta$. Then, the angular change is $\langle \mathbf{w}, \mathbf{w} + \varepsilon \rangle = \langle \mathbf{w}, -\delta \mathbf{w} \rangle = \pi$, which rotates the CAB by 180°. $\square$

In general, the angle $\langle \mathbf{w}, \mathbf{w} + \varepsilon \rangle$ can take arbitrary values in $[0, \pi]$ even for a small perturbation $\varepsilon$. This indicates that CABs are vulnerable to small perturbations when $\mathbf{w}$ has a small norm[1]. This has the implication that even a small gradient noise could destabilize the evolution of ReLU features during stochastic optimization and thus destroy the learning signal for the decision boundaries of the network. Such instability could prevent practitioners from using larger learning rates [18].

It might be tempting to think that WN does not suffer from this issue as it decouples the length and the direction of $\mathbf{w}$ as in Eq (3). However, we show that WN can also be vulnerable to small perturbations.

**Proposition 2.4** (Instability of WN). *A perturbation $(\varepsilon, \varepsilon')$ to $(\mathbf{v}, l)$ under WN (3) can result in an arbitrarily large change in the angular direction of the CAB if $\mathbf{v}$ has a similar magnitude to $\varepsilon$.*

*Proof.* The change in the angular direction of the CAB under WN is given by

$$\left\langle l \frac{\mathbf{v}}{\|\mathbf{v}\|_2}, (l + \varepsilon') \frac{\mathbf{v} + \varepsilon}{\|\mathbf{v} + \varepsilon\|_2} \right\rangle \equiv \arccos \left( \frac{\mathbf{v}^\text{T}(\mathbf{v} + \varepsilon)}{\|\mathbf{v}\|_2 \|\mathbf{v} + \varepsilon\|_2} \right), \tag{8}$$

which has an identical form to Eq (7). Therefore, in the extreme case one similarly construct a small perturbation $\varepsilon = -(1 + \delta) \mathbf{v}$ with an infinitesimal $\delta$, which rotates the CAB by 180°. $\square$

Furthermore, BN can also suffer from this instability issue as it can be viewed as a generalized version of weight normalization.

---

[1] $\|\mathbf{w}\|_2$ is supposed to be small during training as large weights would lead to overfitting, numerical instability and even divergence [18] (e.g., the popular weight decay method explicitly regularizes $\|\mathbf{w}\|_2$ to be close to zero).

**Proposition 2.5** (Instability of BN). *Without loss of generality, assume that the input* $\mathbf{x}$ *has zero mean. A perturbation* $\varepsilon$ *to the weight* $\mathbf{w}$ *under BN (4) can result in an arbitrarily large change in the angular direction of the CAB if* $\mathbf{w}$ *has a similar magnitude to* $\varepsilon$.

*Proof.* Since $\hat{\mathbb{E}}_{\mathbf{x}}[\mathbf{w}^{\mathrm{T}}\mathbf{x}] = \mathbf{w}^{\mathrm{T}}\hat{\mathbb{E}}_{\mathbf{x}}[\mathbf{x}] = \mathbf{0}$ by assumption, Eq (4) can be re-written as

$$\mathrm{BN}(\mathbf{w}^{\mathrm{T}}\mathbf{x} + b) = \gamma\frac{\mathbf{w}^{\mathrm{T}}\mathbf{x}}{\sqrt{\mathbf{w}^{\mathrm{T}}\hat{\boldsymbol{\Sigma}}\,\mathbf{w}}} + \beta = \gamma\left(\frac{\mathbf{w}}{\|\mathbf{w}\|_{\hat{\boldsymbol{\Sigma}}}}\right)^{\mathrm{T}}\mathbf{x} + \beta, \tag{9}$$

where the norm $\|\mathbf{w}\|_{\hat{\boldsymbol{\Sigma}}}$ is with respect to the data covariance matrix $\hat{\boldsymbol{\Sigma}} = \hat{\mathrm{Var}}[\mathbf{x}]$ estimated by mini-batch statistics. It can be seen that Eq (9) has the same form as WN (3) except that WN fixes $\hat{\Sigma} = \mathbf{I}$. Therefore, the instability argument for WN in the proof of Proposition 2.4 also holds for BN. It is worth noting that the same proof technique can be used to show instability for Layer Normalization (LN) [2], which is another normalization technique widely used in Transformers [61], since BN and LN have identical functional form (4) except that the expectation and variance operators in LN are applied to the feature axis rather than the batch axis. $\qquad\square$

Figs 1(b)-1(e) simulate the evolution behaviors of the CABs under SP and WN in $\mathbb{R}^2$, showing that even a small perturbation $\delta$ of magnitude $10^{-3}$ can drastically change the spatial locations of the CABs. More generally, this instability issue exists in many neural network parameterization and normalization techniques that operate in the Cartesian coordinate, since the fundamental issue is that the change in angular direction of the CAB always has the same unstable form as in Eq (7).

## 3 Geometric Parameterization

This section introduces a novel *Geometric Parameterization (GmP)* and demonstrates its nice theoretical property, eliminating the instability issue in common parameterizations and normalizations.

### 3.1 Characteristic Activation Boundary in the Hyperspherical Coordinate System

In a high dimensional input space, most data points live in a thin shell since the volume of a high dimensional space concentrates near its surface [4]. Intuitively, the spatial locations of CABs should be close to the thin shell where most data points live, since this spatial affinity between CABs and data points will introduce ReLU features (non-linearities) at suitable locations in the input space to separate different inputs $\mathbf{x}$ by assigning them different activation values. The use of hyperspherical coordinate enables us to explicitly control the locations of such non-linearities.

**Definition 3.1.** The spatial location of a CAB in the hyperspherical coordinate system is given by

$$\phi(\lambda, \boldsymbol{\theta}) = -\lambda\,\mathbf{u}(\boldsymbol{\theta}), \tag{10}$$

where the radius $\lambda$ corresponds to $b/\|\mathbf{w}\|_2$ in SP, and the unit directional vector $\mathbf{u}(\boldsymbol{\theta})$ corresponds to $\mathbf{w}/\|\mathbf{w}\|_2$ in SP and is determined by the angle $\boldsymbol{\theta} = [\theta_1, \cdots, \theta_{n-1}]^{\mathrm{T}}$:

$$\mathbf{u}(\boldsymbol{\theta}) = \begin{bmatrix} \cos(\theta_1) \\ \sin(\theta_1)\cos(\theta_2) \\ \sin(\theta_1)\sin(\theta_2)\cos(\theta_3) \\ \vdots \\ \sin(\theta_1)\sin(\theta_2)\cdots\sin(\theta_{n-2})\cos(\theta_{n-1}) \\ \sin(\theta_1)\sin(\theta_2)\cdots\sin(\theta_{n-2})\sin(\theta_{n-1}) \end{bmatrix}. \tag{11}$$

$\mathbf{u}(\boldsymbol{\theta}) \in S^{n-1}$ is a unit directional vector on the unit hypersphere $S^{n-1} := \{\mathbf{x} \in \mathbb{R}^n : \|\mathbf{x}\|_2 = 1\}$.

**Definition 3.2.** The CAB in the hyperspherical coordinate system is given by

$$\mathcal{B}(\lambda, \boldsymbol{\theta}) = \{\mathbf{x} \in \mathbb{R}^n : \mathbf{u}(\boldsymbol{\theta})^{\mathrm{T}}\mathbf{x} + \lambda = 0\}. \tag{12}$$

Geometrically speaking, the angle $\boldsymbol{\theta}$ controls the direction of a CAB, while the radius $\lambda$ controls the distance between the origin and the CAB. Calculating the pre-activation of a ReLU unit for an input $\mathbf{x}$ is equivalent to projecting $\mathbf{x}$ onto the unit vector $\mathbf{u}(\boldsymbol{\theta})$ and then adding the radius $\lambda$ to the signed norm of the projected vector. From this perspective, it is clear that a CAB is a set of inputs whose projections over $\mathbf{u}(\boldsymbol{\theta})$ have signed norm $-\lambda$. For this reason, we refer to this radial-angular decomposition in the hyperspherical coordinate system as *Geometric Parameterization* (GmP).

## 3.2 Geometric Parameterization for ReLU Networks

**Definition 3.3** (GmP). A ReLU unit under geometric parameterization (GmP) is given by

$$z = r\,\mathrm{ReLU}(\mathbf{u}(\boldsymbol{\theta})^\mathrm{T}\,\mathbf{x} + \lambda), \tag{13}$$

where $r, \lambda, \boldsymbol{\theta}$ are three learnable parameters. The radial and angular parameters $\lambda$ and $\boldsymbol{\theta} = [\theta_1, \cdots, \theta_{n-1}]$ specify the spatial location of the CAB, while the scaling parameter $r$ controls the scale of the activation. These parameters have $n + 1$ degrees of freedom in total (same as SP).

**Remark 3.4** (Computational cost). Let $n$=fan-in and $m$=fan-out for a layer. GmP needs to compute $2n - 2$ more scalars $[\sin(\theta_1), \cdots, \sin(\theta_{n-1}), \cos(\theta_1), \cdots, \cos(\theta_{n-1})]$ than SP for each of the $m$ neurons, which incur an extra cost of $\mathcal{O}(mn)$ for all neurons in a layer. However, since the cost of computing the affine transformation for each layer is also $\mathcal{O}(mn)$, the total computational cost of GmP remains $\mathcal{O}(mn)$ for each layer, which is identical to SP.

**Remark 3.5** (Layer-size independent parameter initialization). Unlike existing neural network parameterizations which are sensitive to initialization, GmP can work with less carefully chosen initialization schemes independent of the width of the layer, thanks to an invariant property of the hyperspherical coordinate system. To see this, we first consider the distribution of the angular direction of the CAB under SP. Under popular initialization methods such as the Glorot initialization [16] and He initialization [21], each element in the initial weight vector $\mathbf{w}$ under SP is independently and identically sampled from a zero mean Gaussian distribution with a layer-size dependent variance. Interestingly, this always induces a uniform distribution over the unit $n$-sphere for the direction of the CAB, no matter what variance value is used in that Gaussian distribution. This allows us to initialize the angular parameter $\boldsymbol{\theta}$ uniformly at random by sampling from the von Mises–Fisher distribution [63, 65]. The parameter $\lambda$ is initialized to zero due to its connection $\lambda = {}^{b}/{\|\mathbf{w}\|_2}$ to SP and the common practice to set $b = 0$ at initialization. The scaling parameter $r$ is initialized to one, based on the intuition that the scale roughly corresponds to the total variance of the weights $\mathbf{w}$ in SP. We highlight that none of the parameters $\lambda$, $\boldsymbol{\theta}$, and $r$ in GmP requires layer-size dependent initialization.

**Remark 3.6** (Internal covariate shift). One implicit assumption of our characteristic activation analysis is that the input distribution to a neuron always centers around the origin during training. This assumption automatically holds for one-hidden-layer networks since the training data can be centered during data pre-processing. However, this assumption is not necessarily satisfied for multi-hidden-layer networks, since the inputs to an intermediate layer are transformed by the weights and squashed by the activation function in the previous layer. We propose a simple technique called Input Mean Normalization (IMN), which is a *parameter-free* layer that centers the input to each intermediate layer using the empirical mean estimated by mini-batch statistics:

$$z = r\,\mathrm{ReLU}(\mathbf{u}(\boldsymbol{\theta})^\mathrm{T}(\mathbf{x} - \hat{\mathbb{E}}[\mathbf{x}]) + \lambda). \tag{14}$$

It might be tempting to think that IMN is similar to Mean-only Batch Normalization (MBN) [56]. However, we emphasize that MBN is unable to address the internal covariate shift problem as it is applied to pre-activations rather than post-activations.

## 3.3 Theoretical Analysis

Section 2.3 identified an instability in the evolution of CABs under common parameterizations and normalizations due to a fundamental issue in the Cartesian coordinate system. In contrast, CABs under GmP is much more stable under perturbation, since the radius $\lambda$ and angle $\boldsymbol{\theta}$ of the spatial location $\phi$ are automatically disentangled in the hyperspherical coordinate system. This means that small perturbations to the parameters in GmP will only cause small changes in the spatial location of the CAB. Below, we show that under a small perturbation, the change in the angular direction of a CAB under GmP is bounded by the magnitude of the perturbation.

**Theorem 3.7.** *With an infinitesimal perturbation $\boldsymbol{\varepsilon} := [\varepsilon_1, \cdots, \varepsilon_{n-1}]^T$ to the angular parameter $\boldsymbol{\theta}$, the change in the angular direction $\mathbf{u}(\boldsymbol{\theta}) \in S^{n-1}$ $(n \geq 2)$ of a CAB under GmP is bounded by*

$$\langle \mathbf{u}(\boldsymbol{\theta}), \mathbf{u}(\boldsymbol{\theta} + \boldsymbol{\varepsilon}) \rangle \equiv \arccos\big(\mathbf{u}(\boldsymbol{\theta})^\mathrm{T}\mathbf{u}(\boldsymbol{\theta} + \boldsymbol{\varepsilon})\big) = \sqrt{\varepsilon_1^2 + \sum_{i=2}^{n-1}\left(\prod_{j=1}^{i-1}\sin^2(\theta_j)\right)\varepsilon_i^2} \leq \|\boldsymbol{\varepsilon}\|_2. \tag{15}$$

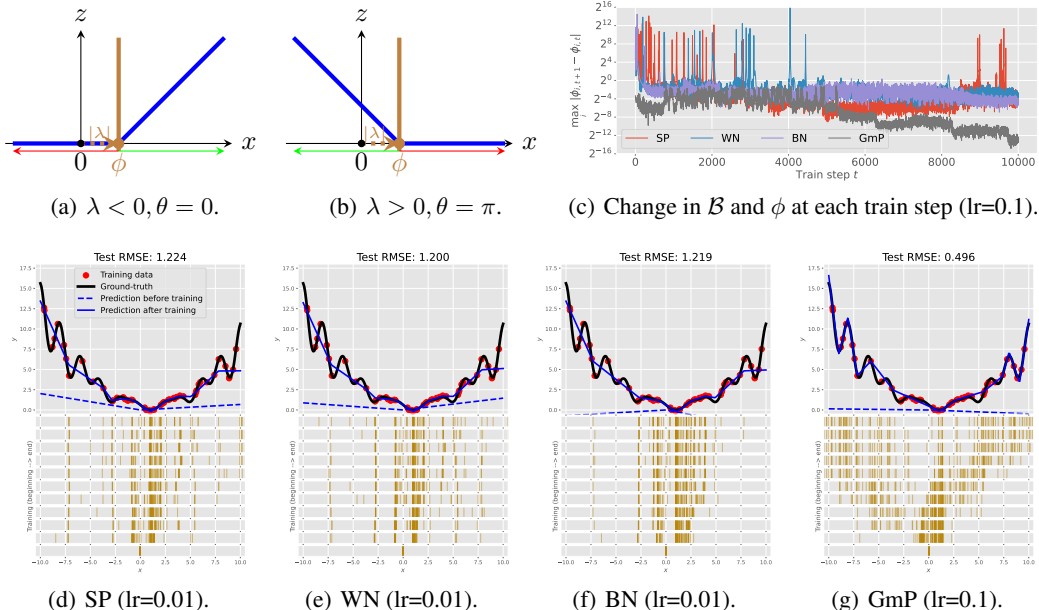

Figure 2: (a)-(b) Characteristic activation point $\mathcal{B}$ (intersection of brown solid lines and the x-axis) and spatial location $\phi = -\lambda u(\theta)$ of a ReLU unit $z = \text{ReLU}(u(\theta)x + \lambda)$ (blue solid lines) for inputs $x \in \mathbb{R}$. Green arrows denote active regions, and red arrows denote inactive regions. (c) Evolution dynamics of the characteristic points $\mathcal{B}$ in a one-hidden-layer network with 100 ReLU units for a 1D Levy regression problem under SP, WN, BN and GmP during training. SP stands for standard parameterization, WN stands for weight normalization, BN stands for batch normalization, and GmP stands for geometric parameterization. Smaller values are better as they indicate higher stability of the evolution of the characteristic points during training. The y-axis is in $\log_2$ scale. (d)-(g): The top row illustrates the experimental setup, including the network's predictions at initialization and after training, and the training data and the ground-truth function (Levy). Bottom row: the evolution of the characteristic activation point for the 100 ReLU units during training. Each horizontal bar shows the spatial location spectrum for a chosen optimization step, moving from the bottom (at initialization) to the top (after training with Adam). More spread of the spatial locations covers the data better and adds more useful non-linearities to the model, making prediction more accurate. Regression accuracy is measured by root mean squared error (RMSE) on a separate test set. Smaller RMSE values are better. We use cross-validation to select the learning rate for each method. The optimal learning rate for SP, WN, and BN is lower than that for GmP, since their training becomes unstable with higher learning rates, as shown in (c).

The proof of Theorem 3.7 can be found in Appendix B, which is based on an elegant idea from differential geometry that the angle $\langle \mathbf{u}(\boldsymbol{\theta}), \mathbf{u}(\boldsymbol{\theta} + \boldsymbol{\varepsilon}) \rangle$ is simply the norm $\|\boldsymbol{\varepsilon}\|_{\mathbf{M}_{\boldsymbol{\theta}}}$ of the perturbation with respect to the metric tensor $\mathbf{M}_{\boldsymbol{\theta}}$ for the hyperspherical coordinate, which is bounded by $\|\boldsymbol{\varepsilon}\|_2$. This implies that optimizing the geometric parameters in GmP directly translates into a smooth evolution of the spatial locations of CABs even in the presence of stochastic gradient noise. Looking back at Figs 1(b)-1(e), we can see that CABs under GmP gradually moves away from its original spatial location as we increase $\delta$, which is in sharp contrast to the unstable evolution of CABs under other parameterizations.

## 4 Empirical Evaluation

This section contains empirical evaluation of GmP with neural network architectures of different sizes on both illustrative demonstrations and more challenging machine learning classification and regression benchmarks. A more detailed setup for each experiment can be found in Appendix C[2].

---

[2]Our code is available at `https://github.com/Wenlin-Chen/geometric-parameterization`.

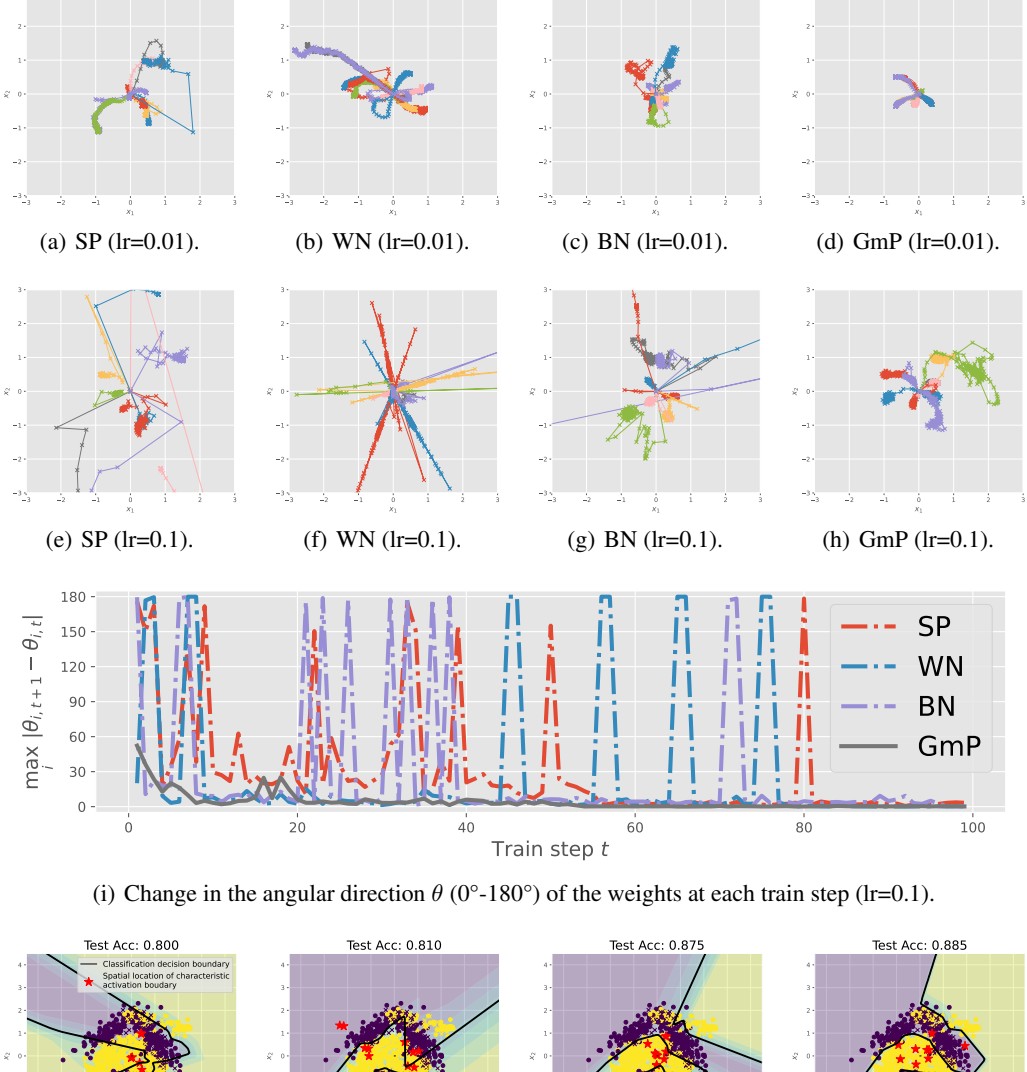

(a) SP (lr=0.01).  (b) WN (lr=0.01).  (c) BN (lr=0.01).  (d) GmP (lr=0.01).

(e) SP (lr=0.1).  (f) WN (lr=0.1).  (g) BN (lr=0.1).  (h) GmP (lr=0.1).

(i) Change in the angular direction $\theta$ (0°-180°) of the weights at each train step (lr=0.1).

(j) SP (lr=0.01).  (k) WN (lr=0.01).  (l) BN (lr=0.01).  (m) GmP (lr=0.1).

Figure 3: Performance of a single-hidden-layer neural network with 10 ReLU units on the 2D Banana classification dataset under SP, WN, BN and GmP trained using Adam. SP stands for standard parameterization, WN stands for weight normalization, BN stands for batch normalization, and GmP stands for geometric parameterization. (a)-(h): Trajectories of the spatial locations of the 10 ReLU units during training. Each color depicts one ReLU unit. Smoother evolution means higher training stability. The evolution under GmP is stable, so we can use a $10\times$ larger learning rate. (i): Evolution dynamics of the angular direction $\theta$ of CABs. Smaller values are better as they indicate higher robustness against stochastic gradient noise. (j)-(m): Network predictions after training. Black bold lines depict the classification boundary between two classes. Classification accuracy is measured on a separate test set. Higher accuracy values are better. The red stars show the spatial locations of 10 ReLU units. Intuitively speaking, more evenly spread out red stars are better for classification accuracy, as they provide more useful non-linearity.

Table 1: Test RMSE for MLPs trained on 7 UCI benchmarks.

| Benchmark | Boston | Concrete | Energy | Naval | Power | Wine | Yacht |
|-----------|--------|----------|--------|-------|-------|------|-------|
| SP | $3.370 \pm 0.145$ | $5.472 \pm 0.144$ | $0.898 \pm 0.274$ | $0.002 \pm 0.000$ | $4.065 \pm 0.029$ | $0.623 \pm 0.008$ | $0.639 \pm 0.063$ |
| WN | $3.459 \pm 0.156$ | $5.952 \pm 0.148$ | $2.093 \pm 0.789$ | $0.003 \pm 0.000$ | $4.073 \pm 0.026$ | $0.632 \pm 0.008$ | $0.624 \pm 0.076$ |
| BN | $3.469 \pm 0.153$ | $5.695 \pm 0.160$ | $1.648 \pm 0.302$ | $\mathbf{0.001 \pm 0.000}$ | $4.164 \pm 0.026$ | $0.622 \pm 0.011$ | $0.777 \pm 0.055$ |
| **GmP** | $\mathbf{3.057 \pm 0.144}$ | $\mathbf{5.153 \pm 0.098}$ | $\mathbf{0.474 \pm 0.013}$ | $0.003 \pm 0.001$ | $\mathbf{4.022 \pm 0.025}$ | $\mathbf{0.613 \pm 0.006}$ | $\mathbf{0.584 \pm 0.046}$ |

## 4.1 Illustrative Experiments

This section verifies the validity of the hypotheses of our proposed characteristic activation analysis on two illustrative experiments aided with visualization, and demonstrates that the improved stability under GmP is beneficial for neural network optimization and generalization.

### 4.1.1 1D Levy Regression

In Fig 2, we train a one-hidden-layer network with 100 ReLU units under SP, WN, BN and GmP on the 1D Levy regression dataset using Adam [28]. As shown in Figs 2(a)-2(b), both the CAB $\mathcal{B}$ and its spatial location $\phi$ reduce to the same point in $\mathbb{R}$, which will be referred to as the characteristic activation point. The angle $\theta$ of the characteristic activation point can only take two values 0 or $\pi$ corresponding to the two directions on the real line. To use GmP in 1D, $\theta$ is initialized to 0 or $\pi$ uniformly at random and fixed throughout training. Clearly, GmP significantly improves the stability of the evolution of the characteristic activation point and allows us to use a $10\times$ large learning rate as selected by cross validation. Fig 2(c) shows that the maximum change $\max_i |\Delta\phi_{i,t}| = \max_i |\phi_{i,t+1} - \phi_{i,t}|$ under GmP is always smaller than one throughout training at each train step $t$, which allows small but consistent updates to be accumulated. In contract, the changes under other parameterizations can be up to $2^{16}$ at some steps. Such abrupt and huge changes of spatial locations make the evolution of the spatial locations inconsistent. Consequently, it is much harder for optimizers to allocate those activations to the suitable locations during training. In addition, many activations are allocated to the regions that are far away from the data region and cannot be seen in Figs 2(d)-2(f), and these activations become completely useless. The stable evolution of the characteristic point under GmP leads to improved optimization stability and generalization performance (i.e., the best test RMSE) on this task, as shown in Figs 2(d)-2(g).

### 4.1.2 2D Banana Classification

In Fig 3, we train a one-hidden-layer network with 10 ReLU units under SP, WN, BN and GmP on the 2D Banana classification dataset using Adam. Figs 3(a)-3(h) show that GmP allows us to use a $10\times$ larger learning rate (as selected by cross validation) while maintaining a smooth evolution of the characteristic activation boundary. Fig 3(i) shows that GmP is the only method that guarantees stable updates for the angular directions of the CAB during training with a large learning rate: under GmP, the maximum change $\max_i |\Delta\theta_{i,t}| = \max_i |\theta_{i,t+1} - \theta_{i,t}|$ at each train step $t$ remains low throughout training, while under other parameterizations the change can be up to $180°$ at some steps. This verifies the hypothesis in our proposed characteristic activation analysis. Figs 3(j)-3(m) show that under GmP, the spatial locations of CABs move towards different directions during training and spread over all training data points in different regions, which forms a classification decision boundary with a reasonable shape that achieves the best generalization performance (i.e., the highest test accuracy) among all compared methods.

## 4.2 Machine Learning Benchmarks

This section evaluate GmP on common machine learning regression and classification benchmarks with a variety of neural network architectures, demonstrating its broad applicability.

### 4.2.1 UCI Regression with MLP

We evaluate GmP on 7 regression problems from the UCI dataset [11]. We train an MLP with one hidden layer and 100 hidden units for 10 different random 80/20 train/test splits. We use the Adam optimizer [28] with cross-validation. We find that the optimal learning rate is 0.1 for GmP and 0.01 for all the other methods. Table 1 shows that for in most cases, GmP consistently achieves the best test RMSE on all benchmarks, significantly outperforming other methods.

Table 2: Top-1 and top-5 validation accuracy (%) for VGG-6 trained on ImageNet32.

| Metric | Top-1 validation accuracy | | | Top-5 validation accuracy | | |
|---|---|---|---|---|---|---|
| Batch size | 256 | 512 | 1024 | 256 | 512 | 1024 |
| SP | $38.31 \pm 0.13$ | $36.99 \pm 0.11$ | $35.02 \pm 0.03$ | $62.48 \pm 0.14$ | $60.71 \pm 0.18$ | $58.14 \pm 0.39$ |
| WN | $39.13 \pm 0.10$ | $37.92 \pm 0.12$ | $36.17 \pm 0.03$ | $63.28 \pm 0.02$ | $61.93 \pm 0.09$ | $60.16 \pm 0.18$ |
| WN+MBN | $42.22 \pm 0.01$ | $40.96 \pm 0.02$ | $39.33 \pm 0.07$ | $66.04 \pm 0.07$ | $65.08 \pm 0.03$ | $63.32 \pm 0.08$ |
| BN | $42.79 \pm 0.03$ | $41.90 \pm 0.19$ | $41.39 \pm 0.02$ | $67.17 \pm 0.08$ | $66.50 \pm 0.25$ | $65.89 \pm 0.06$ |
| **GmP** | $40.76 \pm 0.09$ | $41.65 \pm 0.09$ | $41.29 \pm 0.08$ | $65.08 \pm 0.08$ | $65.76 \pm 0.05$ | $65.49 \pm 0.06$ |
| **GmP+IMN** | $\mathbf{43.14 \pm 0.05}$ | $\mathbf{43.62 \pm 0.08}$ | $\mathbf{42.70 \pm 0.15}$ | $\mathbf{67.36 \pm 0.05}$ | $\mathbf{67.76 \pm 0.09}$ | $\mathbf{66.98 \pm 0.18}$ |

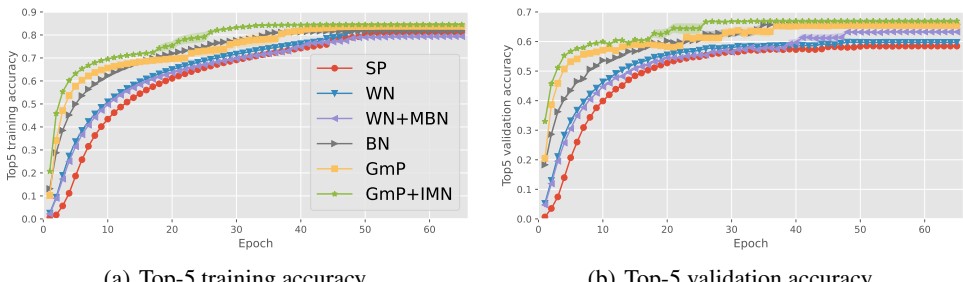

(a) Top-5 training accuracy.  (b) Top-5 validation accuracy.

Figure 4: Convergence speed for VGG-6 trained on the ImageNet32 dataset with batch size 1024.

#### 4.2.2 ImageNet32 Classification with VGG

We evaluate GmP with a medium-sized convolutional neural network VGG-6 [58] on ImageNet32 [8], which contains all 1.3M images and 1,000 categories from ImageNet (ILSVRC 2012) [10], but with the images resized to $32 \times 32$. We follow the experimental setup for optimization and data augmentation as in [8]. This provides insights into how the batch size and intermediate normalization layer affect the empirical convergence speed and generalization performance of different parameterizations. We use cross-validation and find that the optimal initial learning rate is $0.1$ for GmP and $0.01$ for all the other methods. Table 2 shows that GmP+IMN consistently achieves the best top-1 and top-5 validation accuracy for all batch sizes considered. Furthermore, the improvement of GmP+IMN over other methods gets larger as the batch size increases, highlighting the robustness and scalability of GmP with large batch sizes. In addition to achieving the best performance, Fig 4 shows that GmP+IMN (the green curve) also converges significantly faster than other compared methods: its top-5 validation accuracy converges within 25 epochs, which is 10 epochs earlier than the second best method BN. The ablation study GmP vs GmP+IMN shows that IMN significantly improves the performance of GmP, which is expected since it addresses the problem of covariate shifts between hidden layers. Notably, Wide ResNet (WRN 28-2) [68] trained with BN and a batch size of 500 only achieved $43.08\%$ top-1 validation accuracy as reported in [8], underperforming VGG-6 trained with GmP+IMN ($43.62\%$ as shown in Table 2). This reveals the significance of better parameterizations: *even a small non-residual network like VGG-6 with GmP+IMN can outperform large, wide residual networks like WRN 28-2.*

#### 4.2.3 ImageNet Classification with ResNet

We evaluate GmP with a large residual neural network, ResNet-18 [22], on the full ImageNet (ILSVRC 2012) dataset [10], which consists of 1,281,167 training images and 50,000 validation images that contain objects from 1,000 categories. The size of the images ranges from $75 \times 56$ to $4288 \times 2848$. We follow the experimental setup for optimization and data augmentation as in [22]. Specifically, we use the SGD optimizer with momentum 0.9 and reduce the learning rate by $0.1$ at epochs 30, 60 and 80. All models are trained for 90 epochs. We use a batch size of 256 for all methods. We use cross-validation and find that the optimal initial learning

Table 3: Single-center-crop validation accuracy (%) for ResNet-18 trained on ImageNet (ILSVRC 2012).

| Metric | Top-1 valid. acc. | Top-5 valid. acc. |
|---|---|---|
| WN+MBN | $66.57 \pm 0.16$ | $86.69 \pm 0.11$ |
| BN | $66.85 \pm 0.05$ | $86.92 \pm 0.02$ |
| **GmP+IMN** | $\mathbf{67.24 \pm 0.21}$ | $\mathbf{87.19 \pm 0.15}$ |

rate is 0.1 for all compared methods. We employ random horizontal flip, random resizing (256-480) with preserved aspect ratio, random crop (224), and color augmentation for data augmentation during training [31]. To address the internal covariate shift problem, we employ IMN for GmP. Following [56], MBN is used for WN. The result of SP is not reported as BN is used in ResNet by default. Table 3 reports the single-center-crop top-1 and top-5 validation accuracy for all compared methods, which shows that GmP+IMN significantly outperforms BN and WN+MBN in terms of both top-1 and top-5 validation accuracy. This demonstrates that our method is useful for improving large-scale residual network training.

## 5 Related Work

### 5.1 Neural Network Training Dynamics

Neural Tangent Kernels (NTKs) [25, 36] show that wide networks evolve like linear models during training, while Neural Network Gaussian Processes (NNGPs) [49, 35] provide insights into how wide neural networks generalize. [15] studies the evolution of the Hessian spectrum of neural networks during training. [3] investigates the curvatures of different principle components around the optimum of a regularized linear autoencoder. [1, 32, 70] analyze the training dynamics with natural gradient descent (NGD); see Appendix D for a discussion of the connection between GmP and NGD. [45] investigates the effectiveness of stochastic gradient descents for neural network training. [23, 60, 57, 71, 6, 37, 72, 38, 50, 54, 33, 62, 30] study the effects of Adam, BN and weight decay on training dynamics. [64, 26] study the failure cases in joint training of deep ensembles. [7, 53, 34, 5, 43] investigate the training dynamics of distributed optimization with different client subsampling schemes. [48, 52, 13, 14, 19, 20, 12, 55] pay special attention to analyzing the ReLU activations in neural networks. In contrast, our proposed characteristic activation analysis studies the evolution of the characteristic activation boundaries during stochastic optimization.

### 5.2 Neural Network Parameterization and Normalization

In addition to SP, WN [56] and BN [24], there are many other neural network normalization and parameterization techniques. Instead of normalizing the batch axis as in BN, Layer Normalization (LN) [2] operates on the feature axis, which is preferred for small batches or variable-length inputs such as text [61]. Other variants of BN include Switchable Normalization [44] and Instance Enhancement Batch Normalization [39]. There are also normalization techniques designed for specific applications. For instance, Instance Normalization [59] and Group Normalization [66] are special cases of LN designed for CNNs, while Spectral Normalization [47, 69] is specifically designed for GANs and transformers. All these methods operate in the Cartesian coordinate. The instability argument for BN also holds for these methods since they have a similar form to BN except that they normalize the input tensors along different axes. Besides, there is also a line of research on learning orthogonal rotation matrices for the weight vector [42, 40, 41, 51]. In contrast, our proposed Geometric Parameterization (GmP) overcomes the instability issue by operating in the hyperspherical coordinate.

## 6 Conclusion

We presented a novel characteristic activation analysis for understanding the training dynamics of ReLU networks, which exploits special activation values to characterize ReLU units. Using the proposed analysis, we identified a critical instability in common neural network parameterizations and normalization techniques that operate in the Cartesian coordinate. Addressing this, we proposed a new parameterization called Geometric Parameterization (GmP) which operates in the hyperspherical coordinate. We demonstrated the theoretical advantages of GmP for ReLU networks. We performed empirical evaluations to verify our analysis, showing its improved training stability, convergence speed and generalization performance on a variety of tasks with different neural network architectures. More broadly, we believe that the general idea behind our proposed analysis of viewing neural network parameters from a different perspective has the potential to reveal new research directions for the study of neural network optimization and training dynamics. Limitations and potential future work directions are discussed in Appendix E.

## Acknowledgments and Disclosure of Funding

We thank Isaac Reid and Ross Viljoen for helpful feedback and discussions. WC acknowledges funding via a Cambridge Trust Scholarship (supported by the Cambridge Trust) and a Cambridge University Engineering Department Studentship (under grant G105682 NMZR/089 supported by Huawei R&D UK). HG acknowledges generous support from Huawei R&D UK.

Part of this work was performed using resources provided by the Cambridge Service for Data Driven Discovery (CSD3) operated by the University of Cambridge Research Computing Service (`www.csd3.cam.ac.uk`), provided by Dell EMC and Intel using Tier-2 funding from the Engineering and Physical Sciences Research Council (capital grant EP/T022159/1), and DiRAC funding from the Science and Technology Facilities Council (`www.dirac.ac.uk`).

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

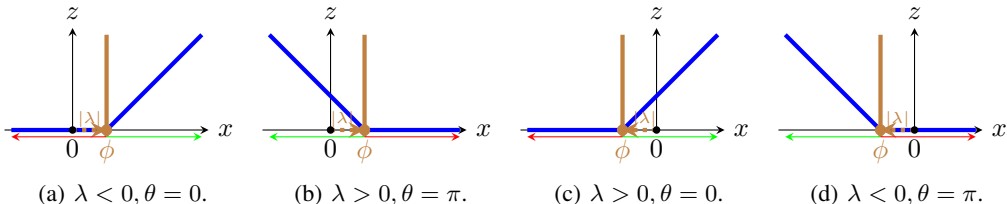

(a) $\lambda < 0, \theta = 0$.  (b) $\lambda > 0, \theta = \pi$.  (c) $\lambda > 0, \theta = 0$.  (d) $\lambda < 0, \theta = \pi$.

Figure 5: Visualization of characteristic activation boundaries (brown solid lines) and spatial locations $\phi = -\lambda u(\theta)$ of a ReLU unit $z = \text{ReLU}(u(\theta)x + \lambda)$ (blue solid lines) for inputs $x \in \mathbb{R}$. Green arrows denote active regions and red arrows denote inactive regions.

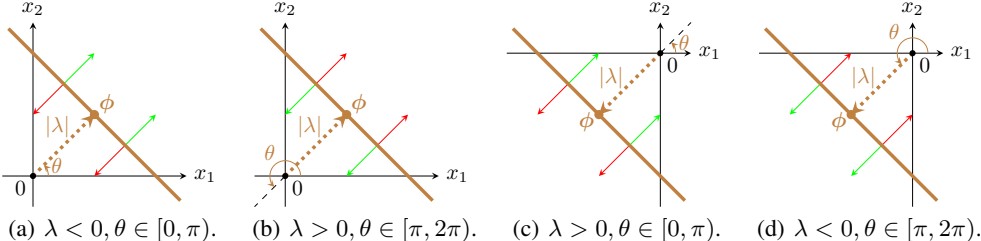

(a) $\lambda < 0, \theta \in [0, \pi)$.  (b) $\lambda > 0, \theta \in [\pi, 2\pi)$.  (c) $\lambda > 0, \theta \in [0, \pi)$.  (d) $\lambda < 0, \theta \in [\pi, 2\pi)$.

Figure 6: Visualization of characteristic activation boundaries (brown solid lines) and spatial locations $\phi = -\lambda\, \mathbf{u}(\theta)$ of a ReLU unit $z = \text{ReLU}(\mathbf{u}(\theta)^\text{T}\, \mathbf{x} + \lambda) = \text{ReLU}(\cos(\theta)x_1 + \sin(\theta)x_2 + \lambda)$ for inputs $\mathbf{x} \in \mathbb{R}^2$. Green arrows denote active regions and red arrows denote inactive regions.

## A  More Visualizations of ReLU Characteristic Activation Boundaries

Figs 5 and 6 respectively visualize the characteristic activation boundaries in the input spaces $\mathbb{R}$ and $\mathbb{R}^2$ under different conditions of the radius and angle parameters.

## B  Proof of Theorem 3.7

*Proof.* Since $\mathbf{u}$ has unit length, the change in the angular direction $\langle \mathbf{u}(\theta), \mathbf{u}(\theta + \varepsilon) \rangle$ under infinitesimal perturbation $\varepsilon$ is equal to the arc length $\delta s$ between the two points $\mathbf{u}(\theta)$ and $\mathbf{u}(\theta + \varepsilon)$. By the generalized Pythagorean theorem, the arc length is given by

$$\delta s = \sqrt{\sum_{i,j} m_{ij}\varepsilon_i\varepsilon_j} = \sqrt{\varepsilon^\text{T}\, \mathbf{M_\theta}\, \varepsilon} = \|\varepsilon\|_{\mathbf{M_\theta}}, \tag{16}$$

where $\mathbf{M_\theta}$ is the metric tensor for the hyperspherical coordinate system. Therefore, the change in the angular direction $\langle \mathbf{u}(\theta), \mathbf{u}(\theta + \varepsilon) \rangle$ under infinitesimal perturbation $\varepsilon$ is simply the norm of $\varepsilon$ with respect to the metric tensor $\mathbf{M_\theta}$ for the hyperspherical coordinate system:

$$\langle \mathbf{u}(\theta), \mathbf{u}(\theta + \varepsilon) \rangle \equiv \arccos\left(\mathbf{u}(\theta)^\text{T}\, \mathbf{u}(\theta + \varepsilon)\right) = \|\varepsilon\|_{\mathbf{M_\theta}}. \tag{17}$$

Hence, we need to work out a formula for calculating $\mathbf{M_\theta}$.

Let $\mathbf{x} = [x_1, \cdots, x_n]^\text{T} \in \mathbb{R}^n$ $(n \geq 2)$ be an input in the Cartesian coordinate system, where the metric tensor is the Kronecker delta $m'_{ij} = \delta_{ij}$. For the Geometric Parameterization of the unit hypersphere $S^{n-1}$, we have

$$u_i(\boldsymbol{\theta}) = \cos(\theta_i) \prod_{k=1}^{i-1} \sin(\theta_k), \quad 0 < i < n,$$

$$u_n(\boldsymbol{\theta}) = \prod_{k=1}^{n-1} \sin(\theta_k). \tag{18}$$

The metric tensor $\mathbf{M}_{\boldsymbol{\theta}}$ for the Geometric Parameterization of $S^{n-1}$ is the pullback of the Euclidean metric in $\mathbb{R}^n$:

$$m_{ab} = \sum_{i=1}^{n} \sum_{j=1}^{n} m'_{ij} \frac{\partial u_i}{\partial \theta_a} \frac{\partial u_j}{\partial \theta_b} = \sum_{i=1}^{n} \frac{\partial u_i}{\partial \theta_a} \frac{\partial u_i}{\partial \theta_b}. \tag{19}$$

We discuss the diagonal ($a = b$) and off-diagonal ($a \neq b$) elements separately:

- $a \neq b$: First, we note that $\frac{\partial u_i}{\partial \theta_q} = 0$ for $0 < i < q$. For $q \leq i \leq n$, we have

$$\frac{\partial u_i}{\partial \theta_q} = -\delta_{iq} \prod_{k=1}^{i} \sin(\theta_k) + \cos(\theta_i) \left( \prod_{k=1}^{i-1} \sin(\theta_k) \right) \left( \sum_{k=1}^{i-1} \frac{\delta_{kq} \cos(\theta_k)}{\sin(\theta_k)} \right), \quad q \leq i < n,$$

$$\frac{\partial u_n}{\partial \theta_q} = \frac{\cos(\theta_q)}{\sin(\theta_q)} \prod_{k=1}^{n-1} \sin(\theta_k). \tag{20}$$

Without loss of generality, we assume $0 < a < b < n$. Then, we have

$$m_{ab} = \sum_{i=1}^{n} \frac{\partial u_i}{\partial \theta_a} \frac{\partial u_i}{\partial \theta_b} = \sum_{i=b}^{n} \frac{\partial u_i}{\partial \theta_a} \frac{\partial u_i}{\partial \theta_b}$$

$$= \sum_{i=b}^{n-1} \cos(\theta_i) \left( \prod_{k=1}^{i-1} \sin(\theta_k) \right) \left( \sum_{k=1}^{i-1} \frac{\delta_{ka} \cos(\theta_k)}{\sin(\theta_k)} \right) \frac{\partial u_i}{\partial \theta_b} + \frac{\cos(\theta_a) \cos(\theta_b)}{\sin(\theta_a) \sin(\theta_b)} \prod_{k=1}^{n-1} \sin^2(\theta_k)$$

$$= \frac{\cos(\theta_a)}{\sin(\theta_a)} \left( \sum_{i=b}^{n-1} \cos(\theta_i) \left( \prod_{k=1}^{i-1} \sin(\theta_k) \right) \frac{\partial u_i}{\partial \theta_b} + \frac{\cos(\theta_b)}{\sin(\theta_b)} \prod_{k=1}^{n-1} \sin^2(\theta_k) \right)$$

$$= \frac{\cos(\theta_a)}{\sin(\theta_a)} \left( -\sin(\theta_b) \cos(\theta_b) \prod_{k=1}^{b-1} \sin^2(\theta_k) \right.$$

$$\left. + \frac{\cos(\theta_b)}{\sin(\theta_b)} \sum_{i=b+1}^{n-1} \cos^2(\theta_i) \left( \prod_{k=1}^{i-1} \sin^2(\theta_k) \right) + \frac{\cos(\theta_b)}{\sin(\theta_b)} \prod_{k=1}^{n-1} \sin^2(\theta_k) \right)$$

$$= \frac{\cos(\theta_a) \cos(\theta_b)}{\sin(\theta_a) \sin(\theta_b)} \left( -\prod_{k=1}^{b} \sin^2(\theta_k) + \sum_{i=b+1}^{n-1} \cos^2(\theta_i) \left( \prod_{k=1}^{i-1} \sin^2(\theta_k) \right) + \prod_{k=1}^{n-1} \sin^2(\theta_k) \right). \tag{21}$$

On the other hand, by recursively collecting like terms and using the identity that $\sin^2(\theta_q) + \cos^2(\theta_q) = 1$, $\forall q$, we have

$$\sum_{i=b+1}^{n-1} \cos^2(\theta_i) \left( \prod_{k=1}^{i-1} \sin^2(\theta_k) \right) + \prod_{k=1}^{n-1} \sin^2(\theta_k)$$

$$= \left( \sum_{i=b+1}^{n-2} \cos^2(\theta_i) \left( \prod_{k=1}^{i-1} \sin^2(\theta_k) \right) + \prod_{k=1}^{n-2} \sin^2(\theta_k) \right) \left( \cos^2(\theta_{n-1}) + \sin^2(\theta_{n-1}) \right)$$

$$= \sum_{i=b+1}^{n-2} \cos^2(\theta_i) \left( \prod_{k=1}^{i-1} \sin^2(\theta_k) \right) + \prod_{k=1}^{n-2} \sin^2(\theta_k) \tag{22}$$

$$= \sum_{i=b+1}^{n-3} \cos^2(\theta_i) \left( \prod_{k=1}^{i-1} \sin^2(\theta_k) \right) + \prod_{k=1}^{n-3} \sin^2(\theta_k)$$

$$= \cdots$$

$$= \prod_{k=1}^{b} \sin^2(\theta_k).$$

This shows that $m_{ab} = 0$ for $a \neq b$ and hence $\mathbf{M}_{\boldsymbol{\theta}}$ is a diagonal matrix.

- $a = b$: Following a similar argument as above, we can obtain the diagonal elements of $\mathbf{M}_{\boldsymbol{\theta}}$:

$$
\begin{aligned}
m_{aa} &= \sum_{i=1}^{n} \left( \frac{\partial u_i}{\partial \theta_a} \right)^2 = \sum_{i=a}^{n} \left( \frac{\partial u_i}{\partial \theta_a} \right)^2 \\
&= \prod_{k=1}^{a} \sin^2(\theta_k) + \sum_{i=a+1}^{n-1} \cos^2(\theta_i) \left( \prod_{k=1}^{i-1} \sin^2(\theta_k) \right) \frac{\cos^2(\theta_a)}{\sin^2(\theta_a)} + \frac{\cos^2(\theta_a)}{\sin^2(\theta_a)} \prod_{k=1}^{n-1} \sin^2(\theta_k) \\
&= \prod_{k=1}^{a} \sin^2(\theta_k) + \frac{\cos^2(\theta_a)}{\sin^2(\theta_a)} \left( \sum_{i=a+1}^{n-1} \cos^2(\theta_i) \left( \prod_{k=1}^{i-1} \sin^2(\theta_k) \right) + \prod_{k=1}^{n-1} \sin^2(\theta_k) \right).
\end{aligned}
\tag{23}
$$

On the other hand, by Eq (22), we have

$$
\sum_{i=a+1}^{n-1} \cos^2(\theta_i) \left( \prod_{k=1}^{i-1} \sin^2(\theta_k) \right) + \prod_{k=1}^{n-1} \sin^2(\theta_k) = \prod_{k=1}^{a} \sin^2(\theta_k).
\tag{24}
$$

Hence, it follows that

$$
\begin{aligned}
m_{aa} &= \prod_{k=1}^{a} \sin^2(\theta_k) + \frac{\cos^2(\theta_a)}{\sin^2(\theta_a)} \prod_{k=1}^{a} \sin^2(\theta_k) \\
&= \sin^2(\theta_a) \prod_{k=1}^{a-1} \sin^2(\theta_k) + \cos^2(\theta_a) \prod_{k=1}^{a-1} \sin^2(\theta_k) \\
&= \left( \sin^2(\theta_a) + \cos^2(\theta_a) \right) \prod_{k=1}^{a-1} \sin^2(\theta_k) \\
&= \prod_{k=1}^{a-1} \sin^2(\theta_k), \quad 2 \le a \le n-1,
\end{aligned}
\tag{25}
$$

and $m_{11} = 1$.

Therefore, the metric tensor for the hyperspherical coordinate is a diagonal matrix

$$
\mathbf{M}_{\boldsymbol{\theta}} =
\begin{bmatrix}
1 & 0 & 0 & \cdots & 0 & 0 \\
0 & \sin^2(\theta_1) & 0 & \cdots & 0 & 0 \\
0 & 0 & \sin^2(\theta_1)\sin^2(\theta_2) & \cdots & 0 & 0 \\
\vdots & \vdots & \vdots & \ddots & \vdots & \vdots \\
0 & 0 & 0 & \cdots & \prod_{i=1}^{n-3} \sin^2(\theta_i) & 0 \\
0 & 0 & 0 & \cdots & 0 & \prod_{i=1}^{n-2} \sin^2(\theta_i)
\end{bmatrix}.
\tag{26}
$$

Finally, the change in the angular direction of a unit vector $\mathbf{u}(\boldsymbol{\theta})$ under a perturbation $\boldsymbol{\varepsilon}$ to $\boldsymbol{\theta}$ is given by the norm of $\boldsymbol{\varepsilon}$ with respect to the tensor matrix $\mathbf{M}_{\boldsymbol{\theta}}$:

$$
\sqrt{\boldsymbol{\varepsilon}^{\mathrm{T}} \mathbf{M}_{\boldsymbol{\theta}} \boldsymbol{\varepsilon}} = \sqrt{\sum_{i=1}^{n-1} m_{ii} \varepsilon_i^2} = \sqrt{\varepsilon_1^2 + \sum_{i=2}^{n-1} \left( \prod_{j=1}^{i-1} \sin^2(\theta_j) \right) \varepsilon_i^2}.
\tag{27}
$$

Furthermore, since $-1 \le \sin(\theta_j) \le 1$, it follows that $0 \le m_{i,i} = \prod_{j=1}^{i-1} \sin^2(\theta_j) \le 1$ for all $i$. Therefore, Eq (27) is bounded by

$$
\sqrt{\varepsilon_1^2 + \sum_{i=2}^{n-1} \left( \prod_{j=1}^{i-1} \sin^2(\theta_j) \right) \varepsilon_i^2} \le \sqrt{\varepsilon_1^2 + \sum_{i=2}^{n-1} \varepsilon_i^2} = \|\boldsymbol{\varepsilon}\|_2.
$$

This completes the proof. $\qquad\square$

## C Detailed Experimental Setups

### C.1 UCI Regression with MLP

We train an MLP with one hidden layer and 100 hidden units for 10 different random 80/20 train/test splits. We use the Adam optimizer [28] with full-batch training. We use cross-validation to select the learning rate for each compared method from the set $\{0.001, 0.003, 0.01, 0.03, 0.1, 0.3\}$. We find that the optimal initial learning rate is $0.1$ for GmP and $0.01$ for all the other compared methods. We report test root mean squared error (RMSE). All models are trained on a single NVIDIA GeForce RTX 2080 Ti.

### C.2 ImageNet Classification with ResNet

We train a ResNet-18 [22] on the ImageNet (ILSVRC 2012) dataset [10], which consists of 1,281,167 training images and 50,000 validation images that contain objects from 1,000 categories. The size of the images ranges from $75 \times 56$ to $4288 \times 2848$. We follow the experimental setup for optimization and data augmentation as in [22]. We use the SGD optimizer with momentum 0.9, which turns out to be better than Adam for image classification tasks [22]. We reduce the learning rate by $0.1$ at epochs 30, 60 and 80. All models are trained for 90 epochs. We use a batch size of 256 for all methods. We use cross-validation to select the learning rate for each compared method from the set $\{0.001, 0.003, 0.01, 0.03, 0.1, 0.3\}$. We find that the optimal initial learning rate is $0.1$ for all compared methods. We employ random horizontal flip, random resizing (256-480) with preserved aspect ratio, random crop (224), and color augmentation for data augmentation during training [31]. To address the internal covariate shift problem, we employ Input Mean Normalization (IMN) for GmP. Following [56], Mean-only Batch Normalization (MBN) is used for WN. We report single-center-crop top-1 and top-5 validation accuracy. The result of SP is not reported as BN is the default normalization for ResNet. All models are trained on a single NVIDIA A100 (80GB).

### C.3 Ablation Study: ImageNet32 Classification with VGG

To maintain a manageable computational cost for the ablation study, we train a VGG-6 [58] on ImageNet32 [8], which contains all 1.3M images and 1,000 categories from ImageNet (ILSVRC 2012) [10], but with images resized to $32 \times 32$. We follow the experimental setup for optimization and data augmentation as in [8]. We use the SGD optimizer with momentum 0.9, which turns out to be better than Adam for image classification tasks [22]. We reduce the learning rate by $0.1$ at epochs 30, 60 and 80. All models are trained for 90 epochs. We train the model using three common batch sizes $\{256, 512, 1024\}$ for all methods. We use cross-validation to select the learning rate for each compared method from the set $\{0.001, 0.003, 0.01, 0.03, 0.1, 0.3\}$. We find that the optimal initial learning rate is $0.1$ for GmP and $0.01$ all the other methods. We employ random horizontal flips for data augmentation during training. We conduct an ablation study to explore the effects of Input Mean Normalization (IMN) for GmP and Mean-only Batch Normalization (MBN) for WN in deep networks. We report top-1 and top-5 validation accuracy. All models are trained on a single NVIDIA GeForce RTX 2080 Ti.

## D Connections to Natural Gradient Descent

The parameter space of a neural network can be thought of as a Riemannian manifold $M$, for which the neural network parameterization specifies the coordinate for $M$. In this paper, we reveal that standard parameterization is vulnerable to small perturbations of the parameters (e.g., SGD noise) whereas our proposed Geometric Parameterization is much more robust against perturbations.

One thing to note is that the full version of natural gradient descent is invariant to neural network parameterizations since it is defined in an abstract form without any specific parameterization/coordinate. In the abstract form, we let the loss function be $l(\theta) = -\log p(Y|X, \theta)$ where $\theta$ are the abstract parameters and $(X, Y)$ is an abstract data point. The fisher matrix $F(\theta) = \mathbb{E}_{p(X,Y)}[\nabla l(\theta)\nabla l(\theta)^T]$ is the metric tensor for $M$, where $p(X, Y)$ is the abstract data distribution. A single step of the full version of natural gradient descent is given by

$$\theta_{t+1} \leftarrow \theta_t - \eta_t \cdot \text{Exp}_{\theta_t}[F(\theta_t)^{-1}\nabla h(\theta_t)], \tag{28}$$

where $h : M \to \mathbb{R}$ is any differentiable function and $\eta_t$ is the learning rate. Note that the exponential map $\text{Exp}_{\theta_t}$ maps the update $F(\theta_t)^{-1}\nabla h(\theta_t)$ from the tangent space $T_{\theta_t}M$ back to the manifold $M$, which results in an exact update invariant to the parameterization/coordinate. However, it is usually infeasible to calculate the exponential map since it requires solving a linear system of size $\dim(M)$, which is the total number of parameters in the neural network. In practice, the most commonly used version of natural gradient descent is its first-order approximation given by

$$\theta_{t+1} \leftarrow \theta_t - \eta_t \cdot F(\theta_t)^{-1}\nabla h(\theta_t), \tag{29}$$

which is a second-order optimization method that is invariant up to first-order transformation of the parameterization/coordinate. Since our proposed Geometric Parameterization is a nonlinear transformation of standard parameterization, it will still behave differently under the first order approximation of the full version of natural gradient descent.

## E    Limitations and Future Work

**Beyond ReLU activation.** This work analyzed ReLU networks due to their wide adoption. However, the proposed characteristic activation analysis is general and can be directly applied to other ReLU-like activation functions such as Leaky ReLU [46], ELU [9] and SReLU [27], because the definition of spatial locations is essentially any breakpoints (i.e., the sources of non-linearity) in a piecewise activation function. The analysis of smooth activations such as Sigmoid and Tanh is left for future work.

**Beyond single-hidden-layer networks.** This work only performed characteristic activation analysis for single-hidden-layer ReLU networks and proposed a practical workaround to address the covariate shift issue between hidden layers by using input mean normalization (IMN). For future work, this analysis needs to be generalized to examine training dynamics in multiple-hidden-layer networks to understand the theoretical behaviors of deep networks. One potential difficulty is that the characteristic activation boundaries of multiple-hidden-layer networks become piecewise linear partitions of the input space, which are less straightforward to analyze. A possible solution would be to consider how the assignment of each data point to the partition evolves during training, similar to how we track the characteristic activation boundaries.

**Sparse activation and large-width limiting behavior.** Interestingly, the product structure of Geometric Parameterization (GmP) as shown in Eq (11) indicates that it could lead to sparse activations or even vanished gradients. Empirically, we find that the activations and gradients of GmP will not vanish during training if the GmP parameters are initialized from the von Mises–Fisher distribution (i.e., uniformly distributed on the hypersphere) as discussed in Remark 3.5. For future work, it would be interesting to investigate the large-width limiting behavior of GmP (e.g., similar to NTK [25] or $\mu$P [67]) to provide theoretical guarantees of the sparsity patterns in GmP activations.

**Efficient representation of hidden neurons.** Directly representing the hidden neurons rather than the parameters in the hyperspherical coordinate could be a more efficient representation. However, it is unclear how to train such neural networks since gradient-based optimization is no longer applicable. One approach would be to use rejection sampling to allocate the hidden neurons on the hypersphere, but it is inefficient and suffers from curse of dimensionality. For future work, it would be interesting to find a more efficient learning algorithm that directly allocates hidden neurons on the hypersphere.

**Combining GmP with existing techniques.** We leave the investigation of the theoretical properties and empirical performance of combinations of GmP and existing normalization (e.g., BN or LN) and regularization techniques (e.g., weight decay) for future work. It would also be interesting to examine the performance of GmP with other neural network architectures (e.g., transformers) on large datasets from different domains (e.g., NLP) under future work.

