# OpenReview forum: "Neural Characteristic Activation Analysis and Geometric Parameterization for ReLU Networks"
_NeurIPS.cc/2024/Conference — NeurIPS 2024 poster_

### Official Review · Reviewer_FLhQ · 2024-07-08

**Soundness:** 3
**Presentation:** 3
**Contribution:** 3
**Rating:** 7
**Confidence:** 4

**Summary:**

This paper identifies and attempts to solve the problem of unstable activation boundaries in ReLU neural networks. The authors define a relu unit in terms of its activation boundary which is the set of inputs that cause the preactivation to be zero. The point on this hyperplane closest to the origin (termed a CAB) is used to track each neuron's activation boundary. The authors show that standard parameterization, weight norm and batch norm all suffer from an instability in the direction of the activation boundary when the weights are small since the direction of the weight can change sign. To address this, they propose to operate in hyperspherical coordinates where the angles $\theta_1, ... , \theta_{N-1}$ and radius $\lambda$ are used to define the hyperplane instead. Taking gradient steps in this geometric parameterization maintains bounded changes in the angles under small learning rate due to a nice metric property of this coordinate system. The

**Strengths:**

This paper identifies an issue with ReLU neurons that can lead to a training instability at large learning rates and attempts to solve the problem with an elegant parameterization motivated by switching to a spherical coordinate system. The computational cost is of the same order as the original matrix computation, so it is still efficient to compute, unlike other approaches such as natural gradient descent. Next, the parameterization is size independent which is a nice property not enjoyed by SP networks. Further, the authors provide many experiments which show that the locations of the activation boundaries are more stable under their proposed parameterization, leading to a speed up in optimization.

**Weaknesses:**

There are some assumptions in the analysis that I was unsure about (such as worst case direction of the perturbations. If instead perturbations were randomly oriented with respect to $w$, does it change the conclusions?). The authors also mention that other normalization solutions suffer from similar problems (like Layernorm etc) but this would be useful to demonstrate, perhaps in the Appendix.

**Questions:**

1. Some of the stability analysis is performed under the worst case assumption that $\epsilon \propto w$. Wouldn't this be unlikely to happen for high dimensional inputs? The main instability is caused when the vectors $w$ change sign/direction under a single step of GD. Could this issue also be solved by initializing the weights in SP to have larger variance relative to the learning rate?
2. Have the authors considered what the structure of the gradients look like in this parameterization? Naively it seems like some of the gradient entries would have very small magnitude in high dimension since they involve products of bounded trigonometric functions. Is there a "limiting" large width description in this parameterization like for NTK parameterization or $\mu$P?

---

> ### Author Rebuttal · Authors · 2024-08-06
>
> Thank you for your insightful comments on our paper! We will address your comments point by point below.
>
> > It would be useful to demonstrate that other normalization solutions suffer from similar problems (like Layernorm etc), perhaps in the Appendix.
>
> Thank you for your suggestion. In the camera-ready version, we will add derivations for these to the Appendix. Here, we take Layernorm as an example and illustrate why it suffers from similar problems. The key of the proof for Layernorm is to realize that the only difference bewteen Layernorm and BN is that they normalize the input tensors along different axes (i.e., BN normalizes inputs across the batch axis and Layernorm normalizes inputs across the feature axis). Therefore, the proof for Layernorm is the same as the proof for BN, except that the expectation and variance operators are now computed along the feature axis.
>
> > If perturbations $\varepsilon$ were randomly oriented with respect to $w$, does it change the conclusions of the analysis? What about high dimensional inputs?
>
> No, the conclusion does not change. This is because the angle between $\varepsilon$ and $w$ can only be between 0 and 180 degrees in any dimensional space. Therefore, even if $\varepsilon$ was randomly oriented with respect to $w$ in high dimensional space, there would still be a 50% chance that the angle between them is greater than 90 degrees, in which case the gradient update will be unstable as the direction of the characteristic boundaries/spatial locations is significantly changed.
>
> > Could this instability issue also be solved by initializing the weights in SP to have larger variance relative to the learning rate?
>
> Unfortunately, no. This is because SP is extremely sensitive to the variance of the initialization distribution as discussed in Remark 3.5 of the paper. If we did not strictly follow the variance as suggested in Glorot/He initialization and used a larger variance instead, the final performance of SP would be very poor.
>
> > Have the authors considered what the structure of the gradients look like in this parameterization?
>
> Thank you for this insightful question! Empirically, we find that the structure of the activations and their gradients under GmP do exhibit some sparsity pattern, and we conjecture that this is one of the reasons why GmP has good generalization. Interestingly, even with the product structure and some sparsity pattern, we find that empirically the activations and gradients will not vanish during training if we initialize the GmP parameters from the von Mises–Fisher distribution (i.e., uniformly distributed on the hypersphere) as discussed in Remark 3.5. We conjecture that GmP may have a large width limiting behavior similar to NTK or $\mu$P, which we are currently investigating in a follow-up project. We will add a discussion on this to the camera-ready version.
>
> We thank the reviewer again for their positive and insightful comments which have helped us improve this work.

---

> > ### Comment · Reviewer_FLhQ · 2024-08-10
> >
> > I thank the authors for their detailed responses to my questions. I will maintain my positive score.

---

### Official Review · Reviewer_x1Bf · 2024-07-11

**Soundness:** 4
**Presentation:** 4
**Contribution:** 4
**Rating:** 8
**Confidence:** 2

**Summary:**

This paper introduces a novel approach for analyzing the training dynamics of ReLU networks by examining the characteristic activation boundaries of individual ReLU neurons. The authors figure out the instability in common neural network parameterization and normalization during stochastic optimization. To address this, the authors propose Geometric Parameterization (GmP), which parameterizes the unit vectors by hyperspherical coordinate representation rather than scaling. The authors conduct experiments to verify the effectiveness of the method.

**Strengths:**

This paper proposes a parameterization method based on spherical coordinate transformation. It is novel to analyze neural networks from a geometric perspective, and the method solves the problem of  instability during stochastic optimization.

Besides, this paper is easy to follow writhing, generally well presented.

**Weaknesses:**

Lack discussions on other activation functions.

**Questions:**

Have the authors explored how GmP performs on other activation functions? Does it also perform better?

Suppose we use GmP on the hidden neurons rather than the weights, like scaling operations projecting $x$ onto a hypersphere. In other words, we use $\theta_1,\cdots,\theta_n$ to represent $x$ by spherical coordinate system. Can the authors give some analysis theoretically or experimentally?

---

> ### Author Rebuttal · Authors · 2024-08-06
>
> Thank you for your positive feedback on our paper! We will address your comments point by point below.
>
> > Discuss GmP on other activation functions.
>
> Our characteristic activation analysis is a very general idea that can be applied to a broader family of activation functions beyond ReLU (e.g., Leaky ReLU, ELU, SReLU, etc) with exactly the same argument. This is because our definition of spatial locations is essentially any breakpoints in a piecewise activation function, since those breakpoints are the sources of nonlinearity. We will add this explanation to the camera-ready version. We leave the characteristic activation analysis for smooth activations for future work since we specifically focus on ReLU-like activations as indicated in the title of our paper.
>
> > What if we use GmP on the hidden neurons rather than the weights?
>
> Thank you for this insightful question! We did consider directly representing the hidden neurons in the hyperspherical coordinate, since this is conceptually a more efficient representation. However, we find that it is very difficult to train such neural networks since gradient-based optimization is no longer applicable. We tried to use rejection sampling to allocate the hidden neurons on the hypersphere, but it is inefficient and suffers from curse of dimensionality in high dimensional space. With that being said, we do think that this could potentially be a promising direction and are currently working on a more efficient learning algorithm for allocating hidden neurons on the hypersphere as a follow-up project. We will add a disucssion of this in the camera-ready version.
>
> We thank the reviewer again for their encouraging and insightful comments which have helped us improve this work.

---

### Official Review · Reviewer_ywDS · 2024-07-13

**Soundness:** 1
**Presentation:** 2
**Contribution:** 2
**Rating:** 3
**Confidence:** 3

**Summary:**

Summary.
Authors consider training  neural nets with Adam after a change of coordinate to spherical coordinate system. They show that in spherical coordinate system the direction of the half spaces which specifies the activeness of the Relus in stable with respect to small changes in the angles, hence the optimization is supposed to be easier. The perform some experiments to see the advantage of using this coordinate system for optimization in practice.

**Strengths:**

This work goes against the conventional practive in deep learning which is running first order method in the standard coordinate system, rather it change to spherical coordinates. They further discuss some advantages of this system for stability of the stability of the activation regions. They conduct various experiments to show this advantage empirically. I think this is an interesting idea; the contribution of the current paper should be assessed in their experiments section, where they show the advantage of this change of coordinates in training. However in the theoretical part there are major concerns that I will mention in the following.

**Weaknesses:**

First the proof of Theorem 3.7. is wrong; I think what the authors meant in the argument of the Theorem is $\|u(\theta) - u(\theta + \epsilon)\|$ rather than their dot product (the dot product claim is obviously wrong since if epsilon goes to zero right hand side goes to zero while left hand side is constant). even with $\|u(\theta) - u(\theta + \epsilon)\|$ instead, the proof in appendix is wrong; the correct formula of length given metric $M_\theta$ on a manifold is: \|u(\theta) - u(\theta + \epsilon)\| = \int_{0}^1 \sqrt{\epsilon^\top M_{\theta + t\epsilon} \epsilon}. The proof can be corrected by estimataing $M_{\theta + t\epsilon}$ with $M_\theta$. Though the final argument is expected to be true, perhaps with different constants.

Second, the paper is highly overclaimed about tjhe theoretical results. The theoretical claim of the paper is only that the normal vector of half space change smoothly if one parameterize in spherical coordinates despite normal coordinates, independent of the training algorithm, while the authors are claiming at the beginning that they analyze the behavior of training algorithms in this case and analyze its advantage over normal coordinates which is not the case.

minor comments:
Line 155 -> typo IMH instead of IMN

Figure 2, (d), ..., (g): how is the fact that activations are more spread out related to the smooth evolution of spatial locations? (the plot with the yellow lines), and these are in 1 and 2D as far as I see, do you have a version of these plots in higher dim? (since that is the authors claim)

**Questions:**

-optimizing in the normal coordinate system with relu activation is known to create sparse features, which can help with generalization, and I am not sure if happens if one optimizes in spherical coordinates. Does the authors see sparsity in their training as well, and if not how do they assess this fact with their claim that spherical coordinate has better generalization?

-In particular authors show in experiments that they can pick a larger learning rate in spherical coordinates, but this fact solely cannot be an indicator of better generalization (even if two algorithms A and B run in the same coordinate system and A picks larger learning rate than B it is still not clear which one generalize better). Can the authors further elaborate on their claim about superiority of spherical coordinates for the generalization of the final network?

**Limitations:**

No theoretical justification for superiority of their method for optimization/generalization.

---

> ### Author Rebuttal · Authors · 2024-08-06
>
> Thank you for your constructive feedback! We believe that your major concern regarding Theorem 3.7 is due to a misunderstanding of our notation. Below, we first address your major concern and then respond to your other comments point by point.
>
> > Clarification of the argument of Theorem 3.7
>
> We believe that there is a misunderstanding of our notation. We would like to clarify that the argument of Theorem 3.7 is regarding the **angle** between the two unit vectors $\mathbf{u}(\boldsymbol{\theta})$ and $\mathbf{u}(\boldsymbol{\theta}+\boldsymbol{\varepsilon})$, rather than the dot product or distance between those two vectors. Therefore, what Theorem 3.7 states is that the angle between $\mathbf{u}(\boldsymbol{\theta})$ and $\mathbf{u}(\boldsymbol{\theta}+\boldsymbol{\varepsilon})$ is bounded by $\lVert\boldsymbol{\varepsilon}\rVert_2$, which goes to zero as $\boldsymbol{\varepsilon}\to 0$. Appendix B along with our response to Reviewer D8e2 provides a detailed proof using techniques from differential geometry, which shows that Theorem 3.7 is indeed correct.
>
> We regret that our notation $\left<\mathbf{u}(\boldsymbol{\theta}),\mathbf{u}(\boldsymbol{\theta}+\boldsymbol{\varepsilon})\right>$ could be misleading as we realized that it can be used to denote dot product in some cases. We thank the reviewer for pointing this out. In the camera-ready version, we will explicitly write down the definition of our notation as follows to avoid any confusion or ambiguity:
> $$\left<\mathbf{u}(\boldsymbol{\theta}),\mathbf{u}(\boldsymbol{\theta}+\boldsymbol{\varepsilon})\right>\equiv\text{arccos}(\mathbf{u}(\boldsymbol{\theta})^\text{T}\mathbf{u}(\boldsymbol{\theta}+\boldsymbol{\varepsilon}))=\sqrt{\varepsilon_1^2+\sum_{i=2}^{n-1}\left(\prod_{j=1}^{i-1}\sin^2(\theta_j)\right)\varepsilon_i^2}\leq\lVert\boldsymbol{\varepsilon}\rVert_2$$
>
> > Clarification of claims about theoretical results
>
> Below, we clarify the contributions of our work and our corresponding claims.
> 1. The first contribution of this work is a novel characteristic activation analysis for ReLU networks, which reveals that traditional NN parameterizations and normalizations in the Cartesian coordinate destablize the evolution of the characteristic boundaries during training. This contribution correpsonds to our claim about analyzing the behavior of traditional NN parameterizations and normalizations during training. We do not claim analyzing NN training algorithms.
> 2. The second contribution of this work is the novel geometric parameterization (GmP), which makes a change of coordinate to train NNs in the hyperspherical coordinate. We provide both theoretical guarantees and empirical results to showcase the advantage of GmP in terms of stability, convergence speed, and generalization. This contribution corresponds to our claim about the advantage of training NNs in the hyperspherical coordinate.
>
> We will make our claims more precise as above in the camera-ready version.
>
> > How is the fact that activations are more spread out related to the smooth evolution of spatial locations?
>
> The more spread-out activations of GmP is one of the positive consequences of smooth evolution of the spatial locations. Figures 2g shows that smooth evolution of spatial locations in GmP allows small and consistent changes to be accumulated, resulting in more spread-out activations eventually. In contrast, Figure 2c shows that the change of spatial locations under other parameterizations can be up to $2^{16}$. Such abrupt and huge changes of spatial locations make the evolution of the spatial locations inconsistent. Consequently, it is much harder for optimizers to allocate those activations to the suitable locations during training. In addition, we would like to point out that some of the activations are even allocated to the regions that are far away from the data region and cannot be seen in Figure 2d-2f, and those activations become completely useless. We will add a discussion on this in the camera-ready version.
>
> > Do you have a version of these plots in higher dim?
>
> It is very hard to visualize the trajectories of characteristic points/boundaries in more than 2D. However, our Theorem 3.7 guarantees smooth evoluation under GmP in any dimensions in theory. We use the plots in 1D and 2D act as proofs of concepts in practice, since those are the only cases that can be visualized.
>
> > Do the authors see sparsity in their training as well?
>
> Thanks for this insightful question! Yes, empirically, we find that GmP also creates sparse features. This can be explained by Equation 11, which shows that GmP can easily create sparse features due to the product structure. We conjecture that this is one of the reasons why GmP provides better generalization and are currently working on a follow-up project to investigate this sparsity behavior and its connection to generalization. We will add a discussion on this in the camera-ready version.
>
> > A larger learning rate in spherical coordinates solely cannot be an indicator of better generalization. Can the authors further elaborate on their claim about superiority of spherical coordinates for the generalization of the final network?
>
> We do not claim a larger learning rate is an indicator of better generalization. What we claim is that a larger learning rate makes convergence faster, as shown in Figure 4. Our claim about GmP's better generalization is empirically supported by GmP's better test performance on a variety of ML tasks with different NN architectures. As mentioned above, we are currently investigating why GmP results in better generalization from a theoretical perspective as a follow-up work (e.g., sparsity and large width limiting behavior like NTK and $\mu$P). We will include this clarification in the camera-ready version.
>
> We thank the reviewer again for their insightful questions and for helping us improve this work. We hope that our response has resolved their concerns and that the reviewer will reconsider their rating.

---

> ### Author Response · Authors · 2024-08-09
> **Thank you for your feedback! Please consider our rebuttal and let us know if your concerns have been addressed.**
>
> We thank the reviewer once again for their effort in the reviewing process. As there are only a few working days left in the discussion period, we would like to ask if our response has addressed the reviewer’s concerns. If so, we kindly invite the reviewer to consider raising their rating. If any concerns remain, we are happy to discuss and clarify them further here.

---

> ### Comment · Reviewer_ywDS · 2024-08-14
> **Response**
>
> Thank you for your additional response.
>
> The proof of Theorem 3.7 is still wrong, do you have a reference for the equation you are claiming in your new response? like I said before the correct formula which I think the authors are interested in using here is the definition of length on manifold which is given by $d(u(\theta), u(\theta + \epsilon)) = \int_{0}^1 \sqrt{\epsilon^\top M_{\theta + t\epsilon} \epsilon}.$
>
> Regarding your response to clarify the contributions in your response above, for the first point, I don't understand the meaning of your sentence "analyzing the behavior of traditional NN parameterizations and normalizations during training." where do you exactly analyze the behavior of Relu neural networks during training? other than that, I acknowledge your argument regarding instability of relu in the normal coordinates, which is INDEPENDENT of the training algorithm.
>
> Regarding your second point of contribution, see my response in the above paragraph. Also again I don't understand what you mean by including "convergence speed, and generalization" analysis as a part of your theoretical contribution. Where do you exactly show theoretical arguments about "convergence speed" or "generalization" of training algorithms?

---

> > ### Author Response · Authors · 2024-08-14
> > **Has our latest response addressed your remaining concerns?**
> >
> > We thank the reviewer once again for their effort in the reviewing process. As there are only a few hours left in the discussion period, we would like to ask if **our latest response regarding the calculation of length on manifold** has addressed the reviewer’s concerns. If so, we kindly invite the reviewer to consider raising their rating. If any concerns remain, we are happy to further discuss and clarify them here.

---

> ### Author Response · Authors · 2024-08-14
> **Response to the addition comments from the reviewer**
>
> Thank you for your response to our rebuttal. We believe that there is still a misunderstanding. Below, we respond to your new comments point by point and hope that this will sufficiently address your remaining concerns.
>
> > ...do you have a reference for the equation you are claiming in your new response?
>
> The first part of the equation that we are claiming in our new response is
> $$\left<\mathbf{u}(\boldsymbol{\theta}),\mathbf{u}(\boldsymbol{\theta}+\boldsymbol{\varepsilon})\right>\equiv\text{arccos}(\mathbf{u}(\boldsymbol{\theta})^\text{T}\mathbf{u}(\boldsymbol{\theta}+\boldsymbol{\varepsilon}))$$
> This means that the angle between two unit vectors is given by the arc cosine of the dot product between the two vectors. This is a basic definition from geometry: since the dot product between two vectors x and y is defined as
> $$x^Ty=||x||||y||\cos\left<x,y\right>$$
> it follows that the angle between those two vectors is given by
> $$\left<x,y\right>\equiv\text{arccos}\left(\frac{x^Ty}{||x||||y||}\right)$$
> In our case, since we are dealing with unit vectors whose norms are one, the denominator becomes one. This shows why the first part is right.
>
> The second part of the equation that we are claiming in our new response is
> $$\text{arccos}(\mathbf{u}(\boldsymbol{\theta})^\text{T}\mathbf{u}(\boldsymbol{\theta}+\boldsymbol{\varepsilon}))=\sqrt{\varepsilon_1^2+\sum_{i=2}^{n-1}\left(\prod_{j=1}^{i-1}\sin^2(\theta_j)\right)\varepsilon_i^2}\leq\lVert\boldsymbol{\varepsilon}\rVert_2$$
> This means that the change in the angles between the two unit vectors goes to zero as epsilon goes to zero. The complete proof of this statement can be found in the Appendix B of our paper. Additional details can be found in our response to Reviewer D8e2, which will be included in the final version of the paper. This is the novel part and is one of the main theoretical contributions of our paper.
>
> >  ...the correct formula which I think the authors are interested in using here is the definition of length on manifold
>
> In our latest response below, we have shown that even if under the definition of the length on manifold mentioned by the reviewer, our statement would still hold true. In our paper, we showed a differential version of this statement rather than the integral version suggested by the reviewer. We thank the reviewer for their suggestions and will add this result regarding the length on manifold under Theorem 3.7 in our paper as a corollary.
>
> First, note that we have shown in the proof of Theorem in Appendix B that
> $$\sqrt{\epsilon^T M_{\theta+t\epsilon}\epsilon}=\sqrt{\varepsilon_1^2+\sum_{i=2}^{n-1}\left(\prod_{j=1}^{i-1}\sin^2(\theta_j+t\epsilon)\right)\varepsilon_i^2}\leq\sqrt{\varepsilon_1^2+\sum_{i=2}^{n-1}\varepsilon_i^2}=\lVert\boldsymbol{\varepsilon}\rVert_2$$
> Then, using the definition of length on the manifold provided by the reviewer, we have
> $$d(u(\theta),u(\theta+\epsilon))\equiv \int_0^1 \sqrt{\epsilon^T M_{\theta+t\epsilon}\epsilon}dt\leq\int_0^1 \lVert\boldsymbol{\varepsilon}\rVert_2 dt = \lVert\boldsymbol{\varepsilon}\rVert_2.$$
>
> Therefore, this distance on the manifold would also go to zero as epsilon goes to zero.
>
> > ...where do you exactly analyze the behavior of Relu neural networks during training?
>
> Sorry for the confusion. In our paper, we do not claim to analyze the behavior of ReLU networks during training. Our claim in the paper is *"we analyze the evolution dynamics of the characteristic activation boundaries in ReLU networks."*, which can be found in Line 22 of our paper. Nevertheless, we thank the reviewer for helping us making our claims more precise and will make sure that this claim is consistently and clearly stated in the final version of the paper.
>
> > ...what you mean by including "convergence speed, and generalization" analysis as a part of your theoretical contribution.
>
> Sorry again for the confusion, but we do not include convergence speed and generalization as part of our theoretical contribution. In our abstract and introduction section, we only claim *"We show theoretically that GmP resolves the aforementioned instability issue."* (Lines 8-9) and *"Our theoretical results show that GmP stabilizes the evolution of characteristic activation boundaries"* (Lines 31-32). As promised in our rebuttal, we will make our claims more precise to clearly state that our theoretical contribution only includes the identification and solution to the instability issue, and the convergence speed and generalization are only supported by empirical evidence. We thank the reviewer for helping us making our claims more precise.
>
> We hope that our response has addressed all your concerns and that the reviewer will reconsider their rating.

---

> ### Author Response · Authors · 2024-08-14
> **Additional theoretical results regarding the calculation of the length on the manifold requested by the reviewer**
>
> We would like to add that even if under the definition of length on the manifold mentioned by the reviewer, our argument would still hold true. We demonstrate this below.
>
> First, note that we have shown in the proof of Theorem 3.7 in Appendix B that
> $$\sqrt{\epsilon^T M_{\theta+t\epsilon}\epsilon}=\sqrt{\varepsilon_1^2+\sum_{i=2}^{n-1}\left(\prod_{j=1}^{i-1}\sin^2(\theta_j+t\epsilon)\right)\varepsilon_i^2}\leq\sqrt{\varepsilon_1^2+\sum_{i=2}^{n-1}\varepsilon_i^2}=\lVert\boldsymbol{\varepsilon}\rVert_2$$
> Then, using the definition of length on the manifold provided by the reviewer, we have
> $$d(u(\theta),u(\theta+\epsilon))\equiv \int_0^1 \sqrt{\epsilon^T M_{\theta+t\epsilon}\epsilon}dt\leq\int_0^1 \lVert\boldsymbol{\varepsilon}\rVert_2 dt = \lVert\boldsymbol{\varepsilon}\rVert_2.$$
>
> Therefore, this distance/length on the manifold would go to zero as epsilon goes to zero. Essentially, Theorem 3.7 is a differential form of the statement, whereas this is an integral form of the statement. We thank the reviewer for this suggestion and will add a corollary for this results under Theorem 3.7 in our paper.

---

### Official Review · Reviewer_D8e2 · 2024-07-20

**Soundness:** 4
**Presentation:** 3
**Contribution:** 3
**Rating:** 6
**Confidence:** 3

**Summary:**

In standard neural networks each neuron, and activation function g performs the following operation on the input $x\in\mathbb{R}^{n}$:
$$z=g\left(w^{t}x+b\right)$$
When usually g in the ReLU activation.

In this work the authors identify a critical instability in many common NN settings, which theoretically destabilizes the evolution of the characteristic boundaries, and empirically impedes fast convergence and hurts generalization performance. To address this, they introduce a novel NN parameterization, named Geometric Parameterization (GmP) which operates in the hyper spherical coordinate. The authors show that this parameterization, both theoretically and empirically, improves optimization stability, convergence speed, and generalization performance.

**Strengths:**

The main strength of this paper (besides the actual methods introduced) is that the results are clear and concise. As a reader, it is relatively easy for me, based on the empirical results, to understand how these methods perform compared to other common methods. I will now explain each dimension.

**Originality**: I think that this paper introduces novel ideas into neural networks (NN). The use of a different coordinate system is not something that I have seen before, and it opens the door to many interesting new approaches to NN.

**Quality**: There is not much to say here, as I feel this is a quality paper. All claims and lemmas are proved as needed.

**Clarity**: The clarity of this paper is good. This mainly stems from the fact that the paper separates the explanation of the problem from the explanation of the solution. Both parts are described using theoretical explanations and empirical/graphical examples.

**Significance**: It is hard to estimate significance, as usually only time will tell how significant the research is, but I feel that this paper will be significant. I believe that the main significance lies not necessarily in the methods introduced, but rather in the idea behind them. The idea of a different perspective on the learned parameters has the potential to lead to more unique and potentially beneficial views.

**Weaknesses:**

I will start with a somewhat nit-picking weakness. The paper uses $\mathcal{B}$ and ϕ in two different contexts. ϕ is used in Definition 2.2 as a vector and in Definition 3.1 as a function. $\mathcal{B}$ is used in Definition 2.2 as a hyperplane and in Definition 3.2 as a function. This isn't accidental, as there is a strong connection between the definitions, but I believe that using slight differences in the notations would be better.

Another weakness is that the article hasn't provided an empirical case where GmP isn't the best option. It is possible that GmP is always the best choice (and if so, great), but usually, there are cases where new techniques aren't the best options. It would have been interesting to see a scenario where GmP isn't the best option and (perhaps under future work) understand why.

**Questions:**

I have few technical questions regarding section B – proof of theorem 3.7:

1. It is not clear to me why equation 16 is true. I would suggest adding an explanation or a reference to one.

2. The analyze of the case where a≠b. It isn't immediately clear from equation 18 why " sum of terms that are either zero or with alternating signs". Maybe it is simple calculus, but I suggest adding additional step to show this claim.

**Limitations:**

Yes, the authors do address the limitations. The main limitation is the input mean normalization that being address in appendix E.

---

> ### Author Rebuttal · Authors · 2024-08-06
>
> Thank you for your valuable feedback on our work! We are encouraged by your positive comments on the originality, quality, clarity, and significance of our paper. Below, we address your concerns point by point.
>
> > Explain the notations $\mathcal{B}$ and $\boldsymbol{\phi}$ in different contexts.
>
> We would like to clarify that $\boldsymbol{\phi}$ is a vector and $\mathcal{B}$ is a hyperplane throughout the paper. Taking $\phi$ as an example, the vector $\boldsymbol{\phi}$ in Definitions 2.1 and 3.1 refers to the same vector in different coordinate systems (i.e., Cartesian and hyperspherical coordinate systems, respectively). We write $\boldsymbol{\phi}$ as a function of $\lambda$ and $\boldsymbol{\theta}$ in Definition 3.1 because we want to emphasize that we have changed to a different coordinate system. In the camera-ready version, we will add a sentence to clarify that Definitions 2.1 and 3.1 define the same vector $\boldsymbol{\phi}$ under different coordinate systems, and we will explicitly write $\boldsymbol{\phi}(\mathbf{w},b)$ as a function of $\mathbf{w}$ and $b$ in Definition 2.1 in order to make the contrast between the two definitions clearer. We will also similarly clarify $\mathcal{B}$ in the camera-ready version.
>
> > Discuss empirical cases where GmP isn't the best option and (perhaps under future work) understand why.
>
> We find that GmP is almost always the best option for MLPs in all cases we have seen. For CNNs, we observe that the benefit of GmP becomes less significant as the network gets more and more overparameterized. We suspect that GmP may not be the best option when the network gets extremely overparameterized. We agree with the reviewer that it would be interesting to understand this behavior under future work, which we are currently working on. In addition, as discussed in Appendix E, it would also be interesting to see whether GmP is the best option for Transformers under future work.
>
> > Explain why Equation 16 is true.
>
> Firstly, note that arc length $=$ angle $\times$ radius. In our case, since the radius of a unit vector is one, the angle between the two unit vectors $\mathbf{u}(\boldsymbol{\theta})$ and $\mathbf{u}(\boldsymbol{\theta}+\boldsymbol{\varepsilon})$ is equal to the arc length $\delta s$ between the two points. By the generalized Pythagorean theorem, the arc length between two points with a small change $\delta\boldsymbol{\theta}$ is given by:
> $$\delta s=\sqrt{\sum_{i,j}m_{ij}\delta\theta_i\delta\theta_j}=\lVert \delta\boldsymbol{\theta}\rVert_M=\lVert \boldsymbol{\varepsilon}\rVert_M$$
> In the camera-ready version, we will add this explanation to the proof of Theorem 3.7 in Appendix B.
>
> > Explain why the sum of terms in Equation 18 cancels out in the case where $a\not=b$.
>
> We first write Equation 17 in a more compact form:
> $$u_i(\boldsymbol{\theta})=\cos\theta_i\prod_{k=1}^{i-1}\sin\theta_k,\quad 0<i<n$$
> $$u_n(\boldsymbol{\theta})=\prod_{k=1}^{n-1}\sin\theta_k$$
> It can be seen that $\frac{\partial u_i}{\partial\theta_q}=0$ for $0<i<q$. For $q\leq i \leq n$, we have
> $$\frac{\partial u_i}{\partial\theta_q}=-\delta_{iq}\prod_{k=1}^{i}\sin\theta_k+\cos\theta_i\left(\prod_{k=1}^{i-1}\sin\theta_k\right)\left(\sum_{k=1}^{i-1}\frac{\delta_{kq}\cos\theta_k}{\sin\theta_k}\right),\quad q\leq i<n$$
> $$\frac{\partial u_n}{\partial\theta_q}=\frac{\cos\theta_q}{\sin\theta_q}\prod_{k=1}^{n-1}\sin\theta_k$$
> Without loss of generality, we assume $a<b$. From Equation 18, we have
> $$m_{ab}=\sum_{i=1}^n\frac{\partial u_i}{\partial\theta_a}\frac{\partial u_i}{\partial\theta_b}=\sum_{i=b}^n\frac{\partial u_i}{\partial\theta_a}\frac{\partial u_i}{\partial\theta_b}=\sum_{i=b}^{n-1}\cos\theta_i\left(\prod_{k=1}^{i-1}\sin\theta_k\right)\left(\sum_{k=1}^{i-1}\frac{\delta_{ka}\cos\theta_k}{\sin\theta_k}\right)\frac{\partial u_i}{\partial\theta_b}+\frac{\cos\theta_a\cos\theta_b}{\sin\theta_a\sin\theta_b}\prod_{k=1}^{n-1}\sin^2\theta_k$$
> $$=\frac{\cos\theta_a}{\sin\theta_a}\left(\sum_{i=b}^{n-1}\cos\theta_i\left(\prod_{k=1}^{i-1}\sin\theta_k\right)\frac{\partial u_i}{\partial\theta_b}+\frac{\cos\theta_b}{\sin\theta_b}\prod_{k=1}^{n-1}\sin^2\theta_k\right)$$
> $$=\frac{\cos\theta_a}{\sin\theta_a}\left(-\sin\theta_b\cos\theta_b\prod_{k=1}^{b-1}\sin^2\theta_k+\frac{\cos\theta_b}{\sin\theta_b}\sum_{i=b+1}^{n-1}\cos^2\theta_i\left(\prod_{k=1}^{i-1}\sin^2\theta_k\right)+\frac{\cos\theta_b}{\sin\theta_b}\prod_{k=1}^{n-1}\sin^2\theta_k\right)$$
> $$=\frac{\cos\theta_a\cos\theta_b}{\sin\theta_a\sin\theta_b}\left(-\prod_{k=1}^b\sin^2\theta_k+\sum_{i=b+1}^{n-1}\cos^2\theta_i\left(\prod_{k=1}^{i-1}\sin^2\theta_k\right)+\prod_{k=1}^{n-1}\sin^2\theta_k\right)$$
> On the other hand, by recursively collecting like terms and applying $\sin^2\theta_q+\cos^2\theta_q=1$, we have
> $$\sum_{i=b+1}^{n-1}\cos^2\theta_i\left(\prod_{k=1}^{i-1}\sin^2\theta_k\right)+\prod_{k=1}^{n-1}\sin^2\theta_k=\sum_{i=b+1}^{n-2}\cos^2\theta_i\left(\prod_{k=1}^{i-1}\sin^2\theta_k\right)+\prod_{k=1}^{n-2}\sin^2\theta_k$$
> $$=\sum_{i=b+1}^{n-3}\cos^2\theta_i\left(\prod_{k=1}^{i-1}\sin^2\theta_k\right)+\prod_{k=1}^{n-3}\sin^2\theta_k=\cdots=\prod_{k=1}^b\sin^2\theta_k$$
> This shows that $m_{ab}=0$ for $a\not=b$. While this derivation only involves collecting like terms, we realized that it might not be that straightforward as it involves many steps. We thank the reviewer for pointing this out. In the camera-ready version, we will add this derivation to the proof of Theorem 3.7 in Appendix B.
>
> We hope that this has sufficiently addressed all your concerns. We thank the reviewer again for their insightful questions and for helping us improve this work.

---

> ### Comment · Reviewer_D8e2 · 2024-08-11
>
> Thank you for the detailed response. As a result I increased my rating by 1

---

### Author Rebuttal · Authors · 2024-08-07

We thank all reviewers for their valuable and insightful comments on our paper and for helping us improve this work. We appreciate that three reviewers support the acceptance of our paper and highlight the novelty, quality, clarity and significance of our proposed characteristic activation analysis and geometric parameterization.

We have addressed each reviewer's concerns separately below their respective review. In particular, we believe that Reviewer ywDS's major concern regarding Theorem 3.7 is due to a misunderstanding of our notation, which we have clarified in our rebuttal to their review. We hope that our responses have sufficiently addressed all reviewers' concerns.

---

### Decision · Program_Chairs · 2024-09-25

**Decision:**

Accept (poster)

**Comment:**

This paper highlights instability phenomena that occur at activation boundaries in ReLU neural networks when weights used in standard parameterizations and normalizations are perturbed. To address these issues, the authors use a hyperspherical coordinate system and demonstrate how it may promote the stability of optimization processes and improves generalization.

The reviewers appreciated the motivating problem presented in the paper, instability phenomena, and found the proposed parameterization to be both interesting and potentially beneficial in practice. Finally, some of the reviewers praised the writing and felt the paper was easy to follow.